# The ZmWAKL–ZmWIK–ZmBLK1–ZmRBOH4 module provides quantitative resistance to gray leaf spot in maize

Tao Zhong [1], Mang Zhu[1], Qianqian Zhang[1], Yan Zhang[2], Suining Deng [1], Chenyu Guo[1], Ling Xu[3], Tingting Liu[4], Yancong Li[4], Yaqi Bi[5], Xingming Fan [5], Peter Balint-Kurti[6] & Mingliang Xu [1] ✉

Gray leaf spot (GLS), caused by the fungal pathogens *Cercospora zeae-maydis* and *Cercospora zeina*, is a major foliar disease of maize worldwide (*Zea mays* L.). Here we demonstrate that *ZmWAKL* encoding cell-wall-associated receptor kinase-like protein is the causative gene at the major quantitative disease resistance locus against GLS. The ZmWAKL[Y] protein, encoded by the resistance allele, can self-associate and interact with a leucine-rich repeat immune-related kinase ZmWIK on the plasma membrane. The ZmWAKL[Y]/ZmWIK receptor complex interacts with and phosphorylates the receptor-like cytoplasmic kinase (RLCK) ZmBLK1, which in turn phosphorylates its downstream NADPH oxidase ZmRBOH4. Upon pathogen infection, ZmWAKL[Y] phosphorylation activity is transiently increased, initiating immune signaling from ZmWAKL[Y], ZmWIK, ZmBLK1 to ZmRBOH4, ultimately triggering a reactive oxygen species burst. Our study thus uncovers the role of the maize ZmWAKL–ZmWIK–ZmBLK1–ZmRBOH4 receptor/signaling/executor module in perceiving the pathogen invasion, transducing immune signals, activating defense responses and conferring increased resistance to GLS.

Plants have evolved multiple, varied signal reception/transduction mechanisms to control cellular functions and coordinate defense responses at the cellular, tissue and organismal levels. The initial perception of pathogen infection is mediated by pattern-recognition receptors (PRRs) at the plasma membrane, which include receptor-like kinases (RLKs) and receptor-like proteins[1,2]. The detection of pathogen-associated molecular patterns and damage-associated molecular patterns by cognate PRRs leads to pattern-triggered immunity (PTI), which includes rapid reactive oxygen species (ROS) production, calcium ($Ca^{2+}$) influx, activation of calcium-dependent and mitogen-activated kinases, changes of immune-related gene expression and, in some cases, localized cell death[3–5]. PTI is vital for preventing infection of most nonadapted microbes and restricting the growth of adapted microbes, termed basal resistance[6].

Cell-wall-associated kinases (WAKs) and WAK-like kinases (WAKLs) represent a unique class of RLKs that are major regulators of fungal disease resistance in plant species. In Arabidopsis, *WAKL22/RFO1* confers resistance to a broad spectrum of *Fusarium* races[7]. The maize

[1]State Key Laboratory of Plant Environmental Resilience/College of Agronomy and Biotechnology/National Maize Improvement Center/Center for Crop Functional Genomics and Molecular Breeding, China Agricultural University, Beijing, P.R. China. [2]Institute of Agricultural Biotechnology, Jilin Academy of Agricultural Sciences, Changchun, P.R. China. [3]State Key Laboratory of Plant Environmental Resilience/College of Biological Sciences, China Agricultural University, Beijing, P.R. China. [4]Baoshan Institute of Agricultural Science, Baoshan, P.R. China. [5]Institute of Food Crops, Yunnan Academy of Agricultural Sciences, Kunming, P.R. China. [6]USDA-ARS Plant Science Research Unit, Raleigh NC and Department of Entomology and Plant Pathology, North Carolina State University, Raleigh, NC, USA. ✉e-mail: mxu@cau.edu.cn

gene *ZmWAK* is induced specifically upon *Sporisorium reilianum* infection in the mesocotyl to inhibit hyphal growth[8]. *ZmWAK-RLK1* confers resistance to northern corn leaf blight caused by *Exserohilum turcicum*[9]. Other WAK genes, such as *Xa4* in rice, *TaWAKL4* and *Snn1* in wheat, also confer resistance to fungal diseases[10–12].

Gray leaf spot (GLS) is a major foliar disease of maize caused by the fungal pathogens *Cercospora zeae-maydis* and *Cercospora zeina*[13]. Since its discovery in the United States in the 1920s, GLS has become a severe global maize disease. GLS was reported to cause substantial yield loss in susceptible cultivars[14]. In the northern United States and Ontario from 2016 to 2019, GLS caused the greatest estimated yield losses of any maize disease[15]. GLS resistance is inherited as a typical quantitative trait, and more than 100 quantitative trait loci (QTLs) have been identified across a range of studies[14,16–19]. So far, *ZmCCoAOMT2* (caffeoyl-CoA *O*-methyltransferase 2) and *ZmMM1* (Mexicana lesion mimic 1) are the only two genes identified to be effective against GLS[20,21].

Here we report the map-based cloning of the GLS quantitative disease resistance (QDR) gene *ZmWAKL* and the elucidation of the signal transduction pathway, which activates defense responses against GLS in maize.

## Fine-mapping of *qRgls1* against GLS

We previously mapped the major QDR locus *qRgls1* to the short arm of chromosome 8 in a segregating population derived from a cross between the highly GLS-resistant inbred line Y32 and GLS-susceptible inbred line Q11. *qRgls1* was fine-mapped to a 1.4-Mb interval. The Y32 allele at *qRgls1* enhanced resistance by 19.7–61.3% compared to the Q11 allele[22]. Here we identified further recombinants in the *qRgls1* region with which we fine-mapped *qRgls1* to a 122-kb interval between markers IDP2 and M2 (B73_RefGen_v4; Extended Data Fig. 1). We allowed five key $BC_8F_7$ recombinants to self-pollinate to develop homozygous lines with or without Y32 segments. The Y32 donor segments resulted in significantly reduced levels of GLS in recombinants II, III and IV, but not in recombinants I and V, further confirming the location of *qRgls1* (Fig. 1a). The Y32 allele at *qRgls1* reduced the GLS disease severity index (DSI) significantly by 14.7–21.6% (Extended Data Fig. 2a).

We selected two homozygous lines derived from recombinant IV, which shared a similar genetic background, but differed at *qRgls1*, named these near-isogenic lines NIL-Y32 (with the Y32 allele) and NIL-Q11 (with the Q11 allele; Fig. 1a). NIL-Y32 was more resistant to *C. zeina* than NIL-Q11 at 44 d postinoculation (dpi; Extended Data Fig. 2b,c). *C. zeae-maydis* and *C. zeina* conidiophores erupt through the stomata during infection[13,23]. Here scanning electron microscopy showed that there was much more profuse conidiation on NIL-Q11 compared to NIL-Y32 (Extended Data Fig. 2d). In several previous studies, GLS resistance has been associated with increased time to flowering[14,24]. These two NILs showed no significant differences in three flowering-related traits when grown under long-day or short-day conditions (Extended Data Fig. 2e,f).

We identified, sequenced and annotated two overlapping bacterial artificial chromosome (BAC) clones from the resistant parental line Y32 encompassing the *qRgls1* region (Supplementary Fig. 1a,b). The *qRgls1* interval was ~122 kb in B73 (B73_RefGen_v4) but only ~60 kb in Y32. The three predicted genes within the Y32 *qRgls1* region were all shared with B73, encoded two cell-wall-associated receptor kinase-like proteins (*Zm00001d008457*, hereinafter referred to as *ZmPR5L*; *Zm00001d008458*, hereinafter referred to as *ZmWAKL*) and a hypothetical protein (*Zm00001d008459*). The B73 *qRgls1* region additionally harbored a gene encoding a leucine-rich repeat RLK (*Zm00001d008460*; Fig. 1a and Supplementary Fig. 1b). We can't detect expressions of either *Zm00001d008459* or *Zm00001d008460* by RT–qPCR; however, expressions of both *ZmPR5L* and *ZmWAKL* were detected, and these genes were considered candidates for *qRgls1* (Supplementary Fig. 1c).

We determined the full-size complementary DNA (cDNA) sequence of *ZmWAKL* by rapid amplification of cDNA ends and amplified the full-size cDNA of *ZmPR5L* by RT–PCR (Supplementary Fig. 1d,e). We designated the Y32 alleles *ZmWAKL^Y* and *ZmPR5L^Y* and the Q11 alleles *ZmWAKL^Q* and *ZmPR5L^Q*. *ZmWAKL^Y* differs from *ZmWAKL^Q* predominantly in the extracellular domain (ECD) with only 62.8% sequence identity. The intracellular domain (ICD) is more conserved with 92.9% sequence identity. Both ZmWAKL isoforms are non-arginine-aspartate (non-RD) kinases comprising the extracellular galacturonan-binding domain (GUB), transmembrane (TM) and cytoplasmic serine/threonine kinase (STK) domains (Extended Data Fig. 3a and Supplementary Fig. 2a). By contrast, ZmPR5L^Y and ZmPR5L^Q shared 98.4% sequence identity (Extended Data Fig. 3b and Supplementary Fig. 2b).

## *ZmWAKL* is the causal gene underlying *qRgls1*

We isolated a 9.2-kb Y32 genomic fragment comprising the native *ZmWAKL^Y* allele for the complementation test (Fig. 1b). We independently obtained five transgenic events in the background of the GLS-susceptible line B73. We self-pollinated three of them to develop homozygous transgenic lines expressing *ZmWAKL^Y* that had significantly higher GLS resistance than B73 (Fig. 1c–e). Notably, the enhanced GLS resistance did not cause substantial changes in plant architecture or flowering time (Supplementary Fig. 3a,b). We backcrossed another two transgenic events to Q11 to produce $T_1BC_1F_1$ and $T_1BC_3F_1$ populations. Transgenic plants consistently exhibited significantly lower DSI than their nontransgenic siblings (Fig. 1f–h). These results indicate that the *ZmWAKL^Y* allele confers resistance to GLS.

We also generated ten and three independent transgenic events overexpressing *ZmWAKL^Y-GFP* (GFP, green fluorescent protein) and *ZmWAKL^Q-GFP* in the B73 background, respectively (Fig. 1i and Extended Data Fig. 4a). *ZmWAKL-GFP* was detectable by immunoblotting with α-GFP antibody (Fig. 1j,m and Extended Data Fig. 4b). We self-pollinated five of the *ZmWAKL^Y-GFP* transgenic events to generate homozygous transgenic lines. All these lines exhibited fewer lesions and significantly reduced GLS scales, but similar morphology and

**Fig. 1 | Map-based cloning of the quantitative GLS-resistance gene at the *qRgls1* locus. a**, Recombinant-derived homozygous lines with/without Y32 segments and their GLS-resistance performance ($n = 1151$). The symbols '+'/'−' indicate the presence/absence of the resistance allele at *qRgls1*, respectively. **b**, Schematic diagram of the complementary pCAMBIA3301-*ZmWAKL^Y* construct. **c–h**, Resistance performance of the complementation transgenic lines in self-pollinated (**c–e**) and $T_1BC_1F_1$ and $T_1BC_3F_1$ backcrossed populations (**f–h**). **c,f**, Relative expression of *ZmWAKL* was determined by RT–qPCR ($n = 3$), and the transgene was confirmed by PCR. **d,e**, GLS symptoms (**d**) and scales (**e**) of B73 and homozygous transgenic lines ($n = 297$). **g,h**, DSI values of transgenic and nontransgenic plants in $T_1BC_1F_1$ ($n = 189$) and $T_1BC_3F_1$ ($n = 304$) segregating populations. **i**, Schematic diagram of the overexpressed Ubipro:*ZmWAKL^Y-GFP* construct. **j–o**, GLS resistance of the overexpression transgenic plants in self-pollinated (**j–l**) and backcrossed populations (**m–o**). **j,m**, Relative expression of *ZmWAKL* was determined by RT–qPCR ($n = 3$), and the transgene was confirmed by immunoblotting with α-GFP antibody. **k,l**, GLS symptoms (**k**) and scales (**l**) of B73 and homozygous transgenic lines ($n = 491$). **n,o**, DSI values of transgenic and nontransgenic plants in $T_1BC_1F_1$ ($n = 469$) and $T_1BC_3F_1$ ($n = 805$) segregating populations. Data are presented as mean values in **g** and **n**; means ± s.e. in **h** and **o**; means ± s.d. in **c**, **f**, **j** and **m**. Data in **a**, **e** and **l** are displayed as box and whisker plots with individual data points, the asterisk denotes the mean and the box limits indicate the interquartile range. In **c**, **e**, **j** and **l**, different lowercase letters indicate a significant difference ($P < 0.05$) based on one-way ANOVA with Tukey's test (**c,j**) or Fisher's LSD test (**e,l**). In **a**, **f**, **g**, **m** and **n**, statistical significance was determined by a two-sided Student's *t* test. Statistical significance was determined by a paired *t* test in **h** and **o**. Scale bars in **a**, **d** and **k**, 15 cm. LB and RB, the left and right T-DNA borders, respectively; NOS, nopaline synthase.

flowering time, compared to B73 (Fig. 1k,l and Supplementary Fig. 3c,d). We backcrossed the remaining five *ZmWAKL^Y-GFP* transgenic events to Q11 to generate $T_1BC_1F_1$ and $T_1BC_3F_1$ populations. Transgenic plants displayed significantly lower DSI than their nontransgenic siblings (Fig. 1n,o). No obvious difference in GLS resistance between overexpression and native *ZmWAKL^Y* alleles suggested that the *ZmWAKL^Y* sequence, rather than its expression level, underpinned the variation in GLS resistance. By contrast, plants overexpressing *ZmWAKL^Q-GFP*

showed no differences in DSI relative to their nontransgenic siblings (Extended Data Fig. 4c,d).

We generated seven independent transgenic events in B73 overexpressing *ZmWAKL^C*, a chimeric gene encoding a protein consisting of the ZmWAKL^Y ECD/TM and ZmWAKL^Q ICD (Extended Data Fig. 4e). We backcrossed these events to Q11 to produce $T_1BC_2F_1$ and $T_1BC_3F_1$ populations. Overall, transgenic plants displayed higher *ZmWAKL* expression levels and lower DSI values compared to their nontransgenic siblings

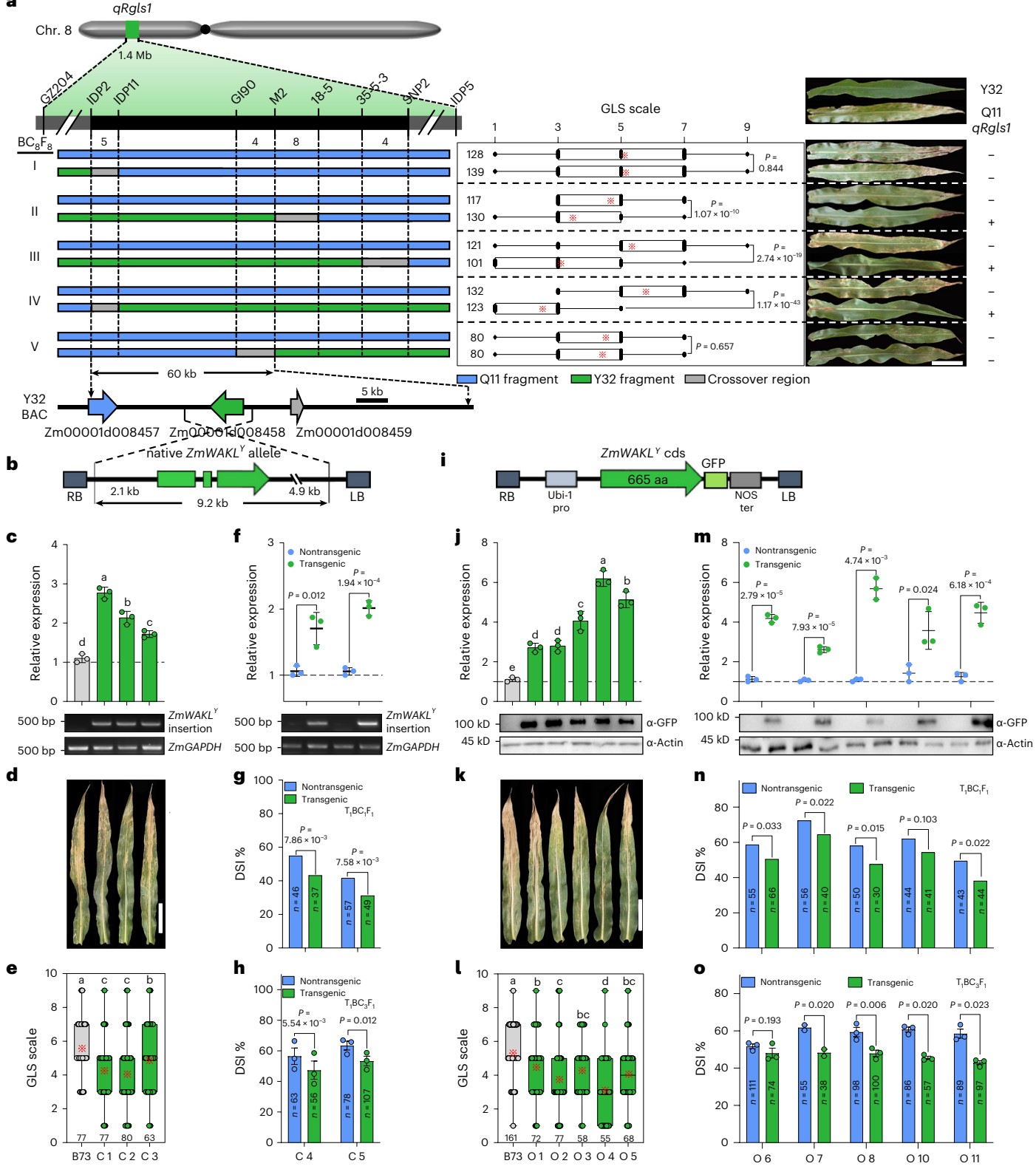

(Extended Data Fig. 4f,g). The chimeric gene *ZmWAKL^C* lowered DSI in Q11 by 4.5–15.6%, a little less than the *ZmWAKL^Y* (7.9–19.5%; Extended Data Fig. 4h), suggesting that the ZmWAKL^Y ECD has a key role in GLS resistance.

For the other candidate, *ZmPR5L*, we obtained three overexpression transgenic events and two clustered regularly interspaced short palindromic repeats (CRISPR)/CRISPR-associated nuclease 9 (Cas9) knocked out *Zmpr5l* null mutants. All transgenic plants showed similar disease severity to B73 (Extended Data Fig. 5), suggesting that *ZmPR5L* is not involved in GLS resistance.

## Molecular characterization of ZmWAKL

The *ZmWAKL^Q* in NIL-Q11 exhibited higher expression levels than the *ZmWAKL^Y* in NIL-Y32 in all tissues except the mesocotyl and root at the seedling stage and the top leaf at the maturity stage (Extended Data Fig. 6a). In plants spray-inoculated at the two-leaf stage in the greenhouse, expression levels of both *ZmWAKL* alleles were rapidly induced by *C. zeina* infection, peaking at 6 h postinoculation (hpi), with *ZmWAKL^Y* expression being higher than *ZmWAKL^Q* (Extended Data Fig. 6b). In field grown plants infusion-inoculated with inocula into the whorl at the 10-leaf stage, the expression levels of both *ZmWAKL^Y* and *ZmWAKL^Q* alleles gradually increased after ~20 dpi and peaked at 29 dpi, with *ZmWAKL^Y* expression higher at the peak (Extended Data Fig. 6c).

We transiently transformed onion epidermal cells with constructs *ZmWAKL^Y-EGFP* (EGFP, enhanced GFP) and *ZmWAKL^Q-EGFP* for localization studies, finding that both ZmWAKL isoforms localized to the plasma membrane (Fig. 2a and Extended Data Fig. 6d). Using both *ZmWAKL^Y-GFP* and *ZmWAKL^Q-GFP* transgenic maize lines, we observed that GFP signal colocalizes with the membrane-impermeable dye FM4-64 (Extended Data Fig. 6e).

Phylogenetic analysis of WAKs/WAKLs reported in rice[10,25–27], tomato[28], Arabidopsis[7,29–32], wheat[11,12,33,34] and maize[8,9] showed them clustered into three clades with a clear distinction between arginine-aspartate (RD) and non-RD kinases. Both ZmWAKL^Y and ZmWAKL^Q are closer to ZmWAK-RLK1 (ref. 9) than to ZmWAK (ref. 8; Supplementary Fig. 4). We surveyed *ZmWAKL* sequences in 98 maize inbred lines (Supplementary Table 2) and identified 48 *ZmWAKL* haplotypes differentiated by numerous SNPs and insertions and deletions (InDels), especially in the extracellular region (Supplementary Fig. 5), which exhibited a nucleotide diversity ($\pi$) >3.4-fold higher than the intracellular region (Supplementary Table 3).

## The self-association and kinase activity of ZmWAKL

Homodimerization occurs as part of the activation process of some PRRs[35]. In a split-ubiquitin-based membrane yeast two-hybrid assay, we observed that the GLS-resistant ZmWAKL^Y, but not the GLS-susceptible ZmWAKL^Q, was able to associate with itself (Fig. 2b). We confirmed these results by split-luciferase complementation (SLC) and co-immunoprecipitation (Co-IP) assays (Fig. 2c,d). Furthermore, we determined that the GUB domain in the ZmWAKL^Y ECD (ZmWAKL^Y/GUB), but not the STK domain in its ICD (ZmWAKL^Y/STK), could self-associate in yeast two-hybrid and SLC assays, while neither ZmWAKL^Q/GUB nor ZmWAKL^Q/STK was able to do so (Fig. 2e,f).

The maltose-binding protein (MBP)-tagged ZmWAKL^Y ICD (ZmWAKL^Y/ICD-MBP), but not its kinase-inactive variant ZmWAKL^Y/ICD,K391E-MBP, exhibited autophosphorylation and phosphorylated the universal kinase substrate myelin basic protein (MBP^a) in vitro, indicating that ZmWAKL^Y/ICD-MBP has kinase activity (Fig. 2g). By contrast, neither ZmWAK^Q/ICD-MBP nor its kinase-inactive variant ZmWAKL^Q/ICD,K430E-MBP had kinase activity (Fig. 2g). We also performed a phosphorylation assay by immunoprecipitating ZmWAKL^Y-GFP and ZmWAKL^Q-GFP from their respective overexpressed transgenic plants. We detected kinase activities from immunoprecipitates of both ZmWAKL-GFP isoforms, as indicated by phosphorylated MBP^a (Fig. 2h). Given that ZmWAKL^Q has no kinase activity in vitro, we speculate that a protein(s) coprecipitated with ZmWAKL^Q-GFP may phosphorylate MBP^a.

## ZmWIK is a coreceptor of ZmWAKL

Plant PRRs usually interact with a coreceptor to recognize the substrate and transduce signal[36]. To identify a possible ZmWAKL coreceptor, we performed repeated IP using extracts from transgenic plants overexpressing *ZmWAKL^Y-GFP* followed by mass spectrometry (MS). Of six identified plasma membrane-tethered kinases (Supplementary Table 4), only a leucine-rich repeat receptor-like serine/threonine-protein kinase (Zm00001d028560, hereinafter referred to as ZmWIK) and a protein kinase (Zm00001d011700, hereinafter referred to as ZmPK2) can interact with both ZmWAKL isoforms (Fig. 3a and Extended Data Fig. 7a–d). ZmPK2, lacking an extracellular receptor-like domain and showing weaker fluorescence and lower gene expression in leaves, is unlikely to be a coreceptor of ZmWAKL (Extended Data Fig. 7a–e).

The localization of a ZmWIK-GFP fusion in onion epidermal cells and *Nicotiana benthamiana* leaves was consistent with the plasma membrane location (Extended Data Fig. 7f,g). In a SLC assay, stronger luminescence was seen in the interaction of *ZmWIK-nLUC* with *cLUC-ZmWAKL^Y* than with *cLUC-ZmWAKL^Q* (Fig. 3b). In a Co-IP assay, ZmWIK-Myc could be coprecipitated with ZmWAKL^Y-GFP and ZmWAKL^Q-GFP (Fig. 3c). To identify the domains of ZmWIK and ZmWAKL responsible for the interaction, we divided ZmWIK and ZmWAKL each into two parts, ECD/TM and ICD. In a SLC assay, ZmWIK^ECD,TM-nLUC interacted with cLUC-ZmWAKL^Y/ECD,TM, but not with cLUC-ZmWAKL^Q/ECD,TM, while ZmWIK^ICD-nLUC interacted relatively weakly with cLUC-ZmWAKL^Y/ICD or cLUC-ZmWAKL^Q/ICD (Fig. 3d). In a pull-down assay, more His-ZmWAKL^Y/ICD was pulled down by ZmWIK^ICD-MBP than His-ZmWAKL^Q/ICD (His, histidine; Fig. 3e). To pinpoint the difference between the two ZmWAKL^ICD isoforms in interacting with ZmWIK, we performed multiple SLC assays. Although cLUC-ZmWAKL^Y/ICD and cLUC-ZmWAKL^Q/ICD showed similar protein abundance, luminescence intensity was significantly higher when *ZmWIK-nLUC* co-expressed with *cLUC-ZmWAKL^Y/ICD* than with *cLUC-ZmWAKL^Q/ICD* (Supplementary Fig. 6). These findings suggest that both the ECD and ICD of ZmWAKL^Y contribute to its preferential interaction with ZmWIK compared to ZmWAKL^Q.

In vitro, the purified His-ZmWIK^ICD exhibited autophosphorylation and strong transphosphorylation of MBP^a, whereas the kinase-inactive variant His-ZmWIK^ICD,K339E did not (Fig. 3f). Although

---

**Fig. 2 | Molecular characterization of ZmWAKL. a**, Subcellular localization of ZmWAKL^Y-EGFP in onion epidermal cells. Scale bars, 50 μm. **b**, Split-ubiquitin-based membrane yeast two-hybrid assay indicates that ZmWAKL^Y could associate with itself. **c**, SLC assay to detect the self-association of two ZmWAKL isoforms. **d**, Co-IP assay to detect the self-association of two ZmWAKL isoforms. The immunoprecipitated products were detected by immunoblotting with α-Myc or α-GFP antibody. **e,f**, Determination of ZmWAKL domains required for self-association in yeast two-hybrid (**e**) and SLC assays (**f**). **g**, Autophosphorylation and kinase activity of the ICDs of two ZmWAKL isoforms. Autoradiography and CBB staining are shown at the top and bottom, respectively. To distinguish

the MBP tag from the MBP substrate in the kinase activity assay, we added a superscript 'a' to the MBP substrate (MBP^a). The experiment was performed three times independently with similar results. **h**, In vitro phosphorylation of MBP^a by ZmWAKL^Y-GFP and ZmWAKL^Q-GFP. ZmWAKL^Y-GFP and ZmWAKL^Q-GFP were separately immunoprecipitated from transgenic plants overexpressing *ZmWAKL^Y-GFP* or *ZmWAKL^Q-GFP*. Phosphorylated MBP^a was detected by autoradiography (autorad), and the protein loading is shown by CBB staining. ZmWAKL-GFP was detected with α-GFP antibody, with plant actin as a loading control. The Co-IP, SLC assays and phosphorylation assays are repeated at least two times.

His-ZmWIK^{ICD} had strong kinase activity, it only weakly phosphorylated the two kinase-inactive ZmWAKL^{ICD}-MBP variants (Fig. 3g). ZmWAKL^{Y/ICD}-MBP also showed autophosphorylation but weak transphosphorylation of His-ZmWIK^{ICD,K339E}, while ZmWAKL^{Q/ICD}-MBP displayed neither autophosphorylation nor transphosphorylation

(Fig. 3h). We speculate that the use of kinase-inactive variants as substrates may underlie the relatively weak reciprocal phosphorylation, like what has been observed in the previous report[37]. When ZmWAKL^{Y/ICD}-MBP and His-ZmWIK^{ICD} were mixed in vitro, they enhanced each other's phosphorylation levels, with His-ZmWIK^{ICD} being more

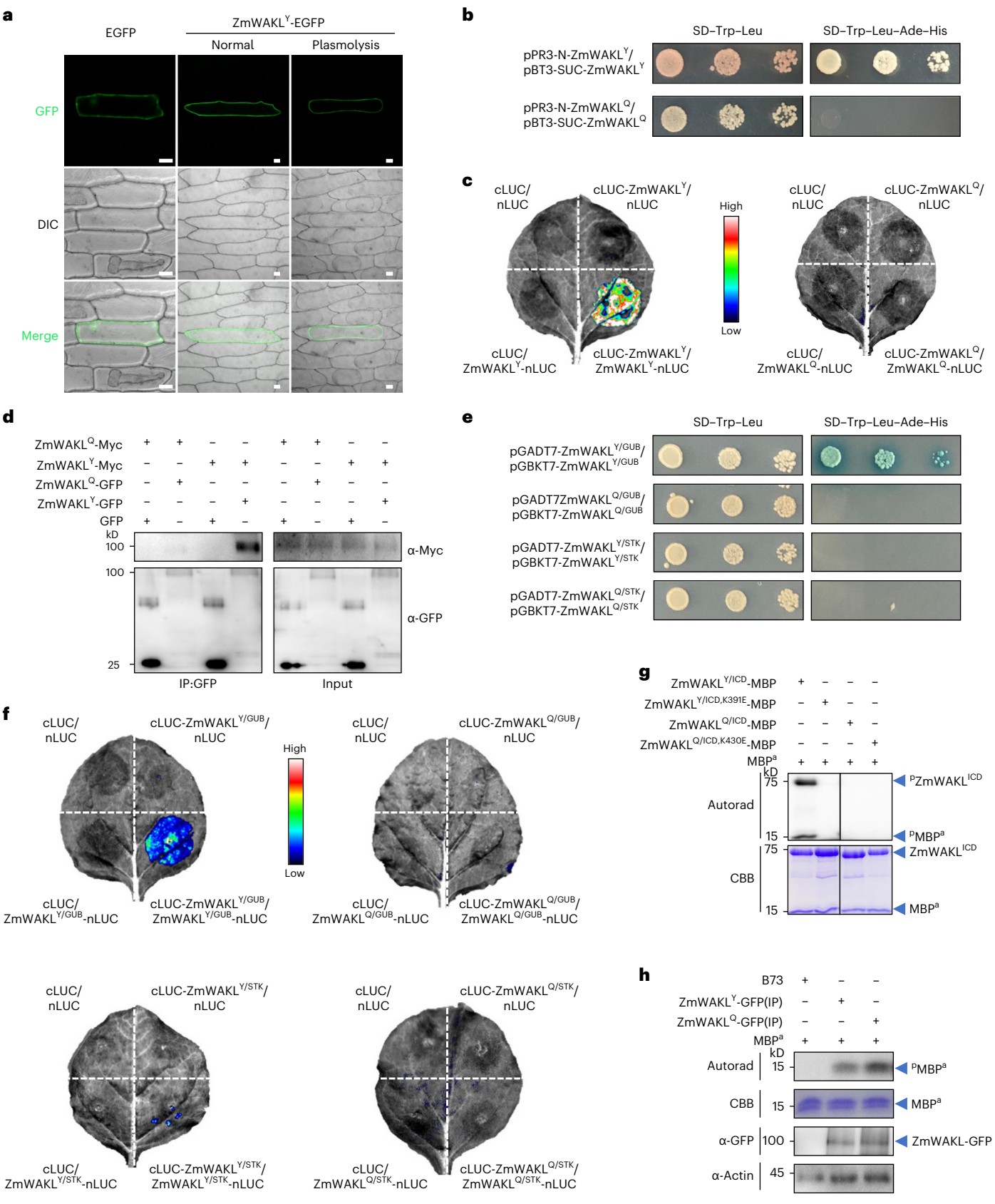

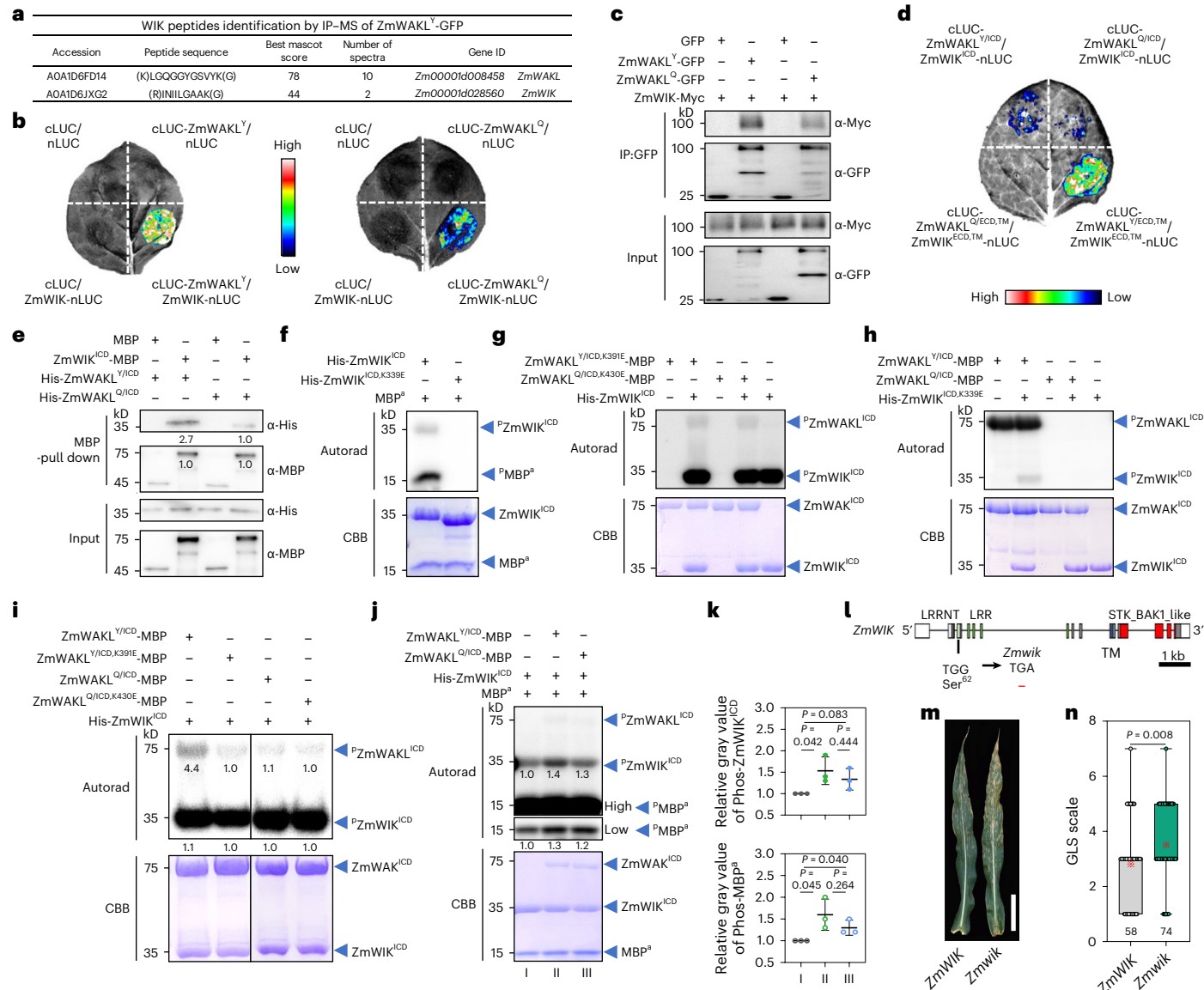

**Fig. 3 | ZmWIK is a coreceptor of ZmWAKL and positively regulates GLS resistance. a**, Identification of ZmWAKL and its interacting RLK ZmWIK by IP–MS analysis. **b,c**, Interaction of ZmWIK with ZmWAKL^Y or ZmWAKL^Q in SLC (**b**) and Co-IP assay (**c**). **d**, Determination of domains responsible for the interactions between ZmWIK and ZmWAKL in a SLC assay. **e**, Interaction of ZmWIK^ICD with ZmWAKL^Y/ICD or ZmWAKL^Q/ICD in an in vitro pull-down assay. **f**, Autophosphorylation and kinase activity of ZmWIK^ICD. **g–i**, Reciprocal phosphorylation between ZmWIK and ZmWAKL in vitro. **g**, Phosphorylation of the kinase-inactive variants ZmWAKL^Y/ICD,K391E-MBP and ZmWAKL^Q/ICD,K430E-MBP by His-ZmWIK^ICD. **h**, Phosphorylation of the kinase-inactive variant His-ZmWIK^ICD,K339E by ZmWAKL^Y/ICD-MBP or ZmWAKL^Q/ICD-MBP. **i**, Reciprocal phosphorylation between His-ZmWIK^ICD and ZmWAKL^Y/ICD-MBP (or ZmWAKL^Q/ICD-MBP or their two kinase-inactive variants). **j,k**, Effects of ZmWAKL^Y/ICD (or ZmWAKL^Q/ICD) on phosphorylation and kinase activity of ZmWIK^ICD (**j**), and the quantification of His-ZmWIK^ICD and MBP^a phosphorylation levels (**k**; $n = 3$ independent experiments). High, high exposure; Low, low exposure. **l**, Schematic diagram of *ZmWIK* and EMS-induced stop codon to generate *ZmWIK* null mutant. **m,n**, GLS resistance in the wild-type *ZmWIK* and *ZmWIK* null mutant ($n = 132$). Scale bar, 15 cm. In **f–j**, protein phosphorylation was detected by autoradiography (upper), and equal protein loading is shown by CBB staining (lower). In **k** and **n**, statistical significance was determined by a two-sided Student's *t* test. In **k**, data are shown as means ± s.d. In **n**, data are displayed as box and whisker plots with individual data points, the dotted cross denotes the mean and the box limits indicate the interquartile range. The Co-IP, SLC, pull-down and phosphorylation assays were repeated at least two times.

strongly phosphorylated than ZmWAKL^Y/ICD-MBP. Reduced phosphorylation levels were observed when using ZmWAKL^Q/ICD-MBP or two kinase-inactive ZmWAKL^ICD-MBP variants in place of ZmWAKL^Y/ICD-MBP (Fig. 3i). His-ZmWIK^ICD strongly phosphorylated MBP^a and adding ZmWAKL^Y/ICD-MBP further increased the phosphorylation levels of both His-ZmWIK^ICD and MBP^a. Intriguingly, ZmWAKL^Q/ICD-MBP had similar effects, although to a lesser extent (Fig. 3j,k).

Two transgenic lines overexpressing *ZmWIK* showed significantly fewer GLS symptoms compared to B73 (Extended Data Fig. 7h–j).

An ethyl methanesulfonate (EMS) induced null mutant *Zmwik*, harboring a premature translation termination codon, was backcrossed into NIL-Y32. The plants with *Zmwik* were more susceptible to GLS than wild-type counterparts in the field trials (Fig. 3l–n), suggesting that *ZmWIK* may act downstream of *ZmWAKL^Y* in GLS resistance. Overall, these data indicate that ZmWIK binds to ZmWAKL^Y at the plasma membrane and lack of either protein reduces GLS resistance. Moreover, they mutually phosphorylate each other, with more frequent phosphorylation made by ZmWAKL^Y to ZmWIK. We thus speculate that ZmWIK may be a coreceptor of ZmWAKL.

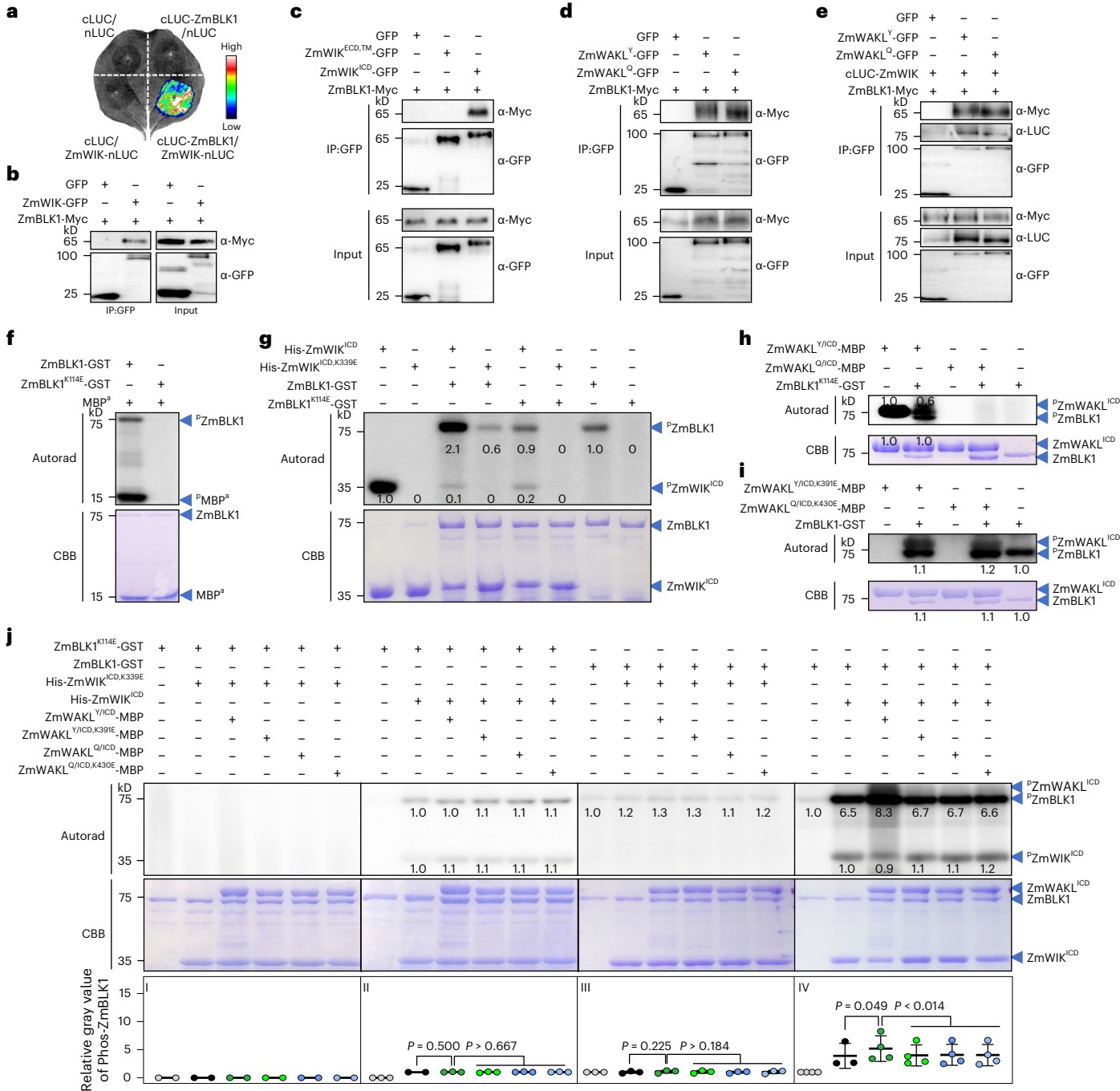

**Fig. 4 | ZmBLK1 is a cytoplasmic kinase downstream of the ZmWAKL/ZmWIK complex. a**, Interaction between ZmBLK1 and ZmWIK in an SLC assay. **b–d**, In Co-IP assay, ZmBLK1 interacts with ZmWIK (**b**), ZmWIK$^{ICD}$ (**c**) and ZmWAKL$^Y$/ZmWAKL$^Q$ (**d**), but not with ZmWIK$^{ECD,TM}$ (**c**). **e**, Interaction among ZmWIK, ZmBLK1 and ZmWAKL$^Y$ (or ZmWAKL$^Q$) in Co-IP assay. **f**, Autophosphorylation and kinase activity of ZmBLK1. **g**, Reciprocal phosphorylation between ZmBLK1 and ZmWIK. His-ZmWIK$^{ICD}$ and ZmBLK1-GST and their kinase-inactive variants His-ZmWIK$^{ICD,K339E}$ and ZmBLK1$^{K114E}$-GST were used for in vitro phosphorylation assays. **h,i**, Reciprocal phosphorylation between ZmBLK1 and ZmWAKL. The in vitro phosphorylation assays were performed using two ZmWAKL$^{ICD}$-MBP isoforms with the kinase-inactive ZmBLK1$^{K114E}$-GST (**h**) and two kinase-inactive

ZmWAKL$^{ICD}$-MBP variants with ZmBLK1-GST (**i**). **j**, Mutual phosphorylation assays among ZmWAKL, ZmWIK and ZmBLK1. Phosphorylation levels of His-ZmWIK$^{ICD,K339E}$ and ZmBLK1$^{K114E}$-GST (I; $n = 2$ independent experiments), His-ZmWIK$^{ICD}$ and ZmBLK1$^{K114E}$-GST (II; $n = 3$ independent experiments), His-ZmWIK$^{ICD,K339E}$ and ZmBLK1-GST (III; $n = 3$ independent experiments) and His-ZmWIK$^{ICD}$ and ZmBLK1-GST (IV; $n = 4$ independent experiments) in the presence of ZmWAKL$^Y$ or ZmWAKL$^Q$ or their kinase-inactive variants. In **f–j**, autoradiograph (upper) and CBB staining (lower) show phosphorylation and loading, respectively. In **j**, the data were shown as means ± s.d., and the statistical significance was determined by a paired $t$ test. The Co-IP, SLC and phosphorylation assays were repeated at least two times.

## ZmWAKL/ZmWIK interacts with ZmBLK1 for immune signaling

RLCKs, Botrytis-induced kinase 1 (BIK1) and PBL1 (avrPphB susceptible 1-like 1), have emerged as core components linking PRRs to downstream

defenses[38]. We hypothesized that a similar RLCK might contribute to ZmWAKL-mediated immunity. Thus, two maize BIK1/PBL1 homologs, ZmBLK1 (Zm00001d034662) and ZmBLK1-1 (Zm00001d012958), sharing 61.1%/69.0% and 60.4%/69.0% identical sequences to AtBIK1

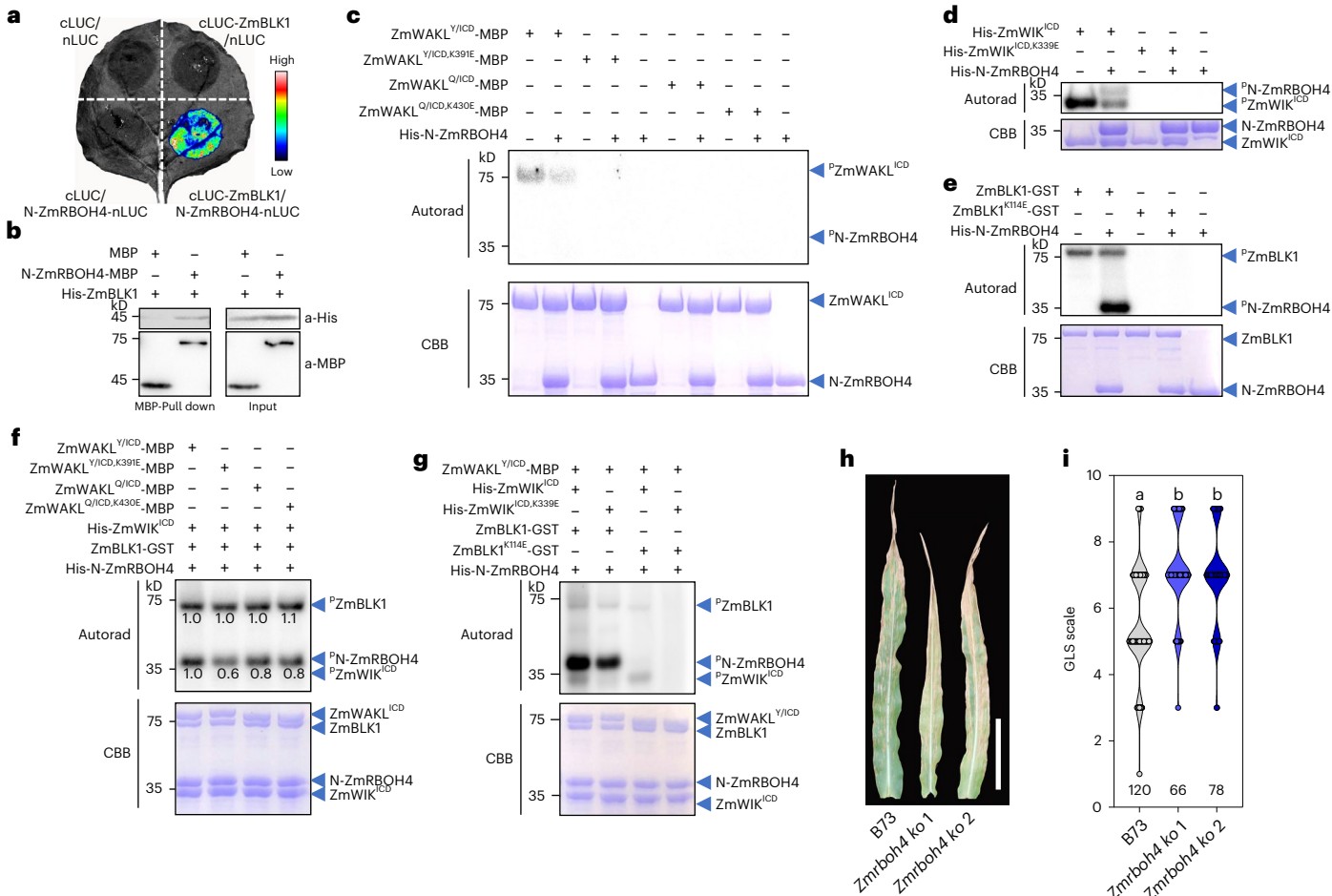

**Fig. 5 | ZmRBOH4 receives immune signal from ZmBLK1 and regulates GLS resistance in maize. a**, N-ZmRBOH4 interacts with ZmBLK1 in SLC assay. **b**, In vitro MBP pull-down assay to show the interaction of His-ZmBLK1 with N-ZmRBOH4-MBP. **c**, ZmWAKL does not phosphorylate N-ZmRBOH4 in an in vitro kinase assay. **d**, ZmWIK weakly phosphorylates N-ZmRBOH4 in vitro. **e**, ZmBLK1 strongly phosphorylates N-ZmRBOH4 in vitro. **f**, Phosphorylation levels of ZmWAKL[ICD], ZmWIK[ICD], ZmBLK1 and N-ZmRBOH4 when mixed in vitro. ZmWAKL[Y/ICD]-MBP, ZmWAKL[Q/ICD]-MBP and their corresponding kinase-inactive variants were separately added to a mixture of His-ZmWIK[ICD], ZmBLK1-GST and His-N-ZmRBOH4 to detect phosphorylation levels. **g**, Effects of the kinase-inactive variants His-ZmWIK[ICD,K339E] and/or ZmBLK1[K114E]-GST on phosphorylation of N-ZmRBOH4. His-ZmWIK[ICD]/ZmBLK1-GST or His-ZmWIK[ICD,K339E]/ZmBLK1-GST

or His-ZmWIK[ICD]/ZmBLK1[K114E]-GST or His-ZmWIK[ICD,K339E]/ZmBLK1[K114E]-GST were separately added to a mixture of ZmWAKL[Y/ICD]-MBP and His-N-ZmRBOH4. **h,i**, Representative images of ear leaves (**h**) and GLS scale values (**i**) of B73 and *ZmRBOH4* knockout plants at 44 dpi with *C. zeina* in the field. Each value represents the disease scale from one plant. The number of plants is indicated below. Scale bar, 15 cm. In **c–g**, protein phosphorylation was detected by autoradiography (upper), and equal protein loading is shown by CBB staining (lower). In **i**, data are shown as violin plots with individual data points; different lowercase letters indicate a significant difference (*P* < 0.05) based on one-way ANOVA with the LSD test. The SLC, pull-down and phosphorylation assays were repeated at least two times.

and AtPBL1, respectively, were identified (Supplementary Fig. 7a). Phylogenetic analysis indicated that ZmBLK1/ZmBLK1-1 clustered in the same clade as OsRLCK176 (refs. 39,40), OsRLCK57 (ref. 41), SlTPK1b (tomato protein kinase 1)[42], AtBIK1 (ref. 43) and AtPBL1 (ref. 44), all of which are involved in ROS production via the NADPH oxidase respiratory burst oxidase homolog D (RBOHD). Other RLCKs, OsRLCK185 (ref. 45), AtPBL27 (PBS-like 27)[46], AtPCRK1/2 (PTI compromised RLCK 1/2)[47] and AtBSK1 (brassinosteroid-signaling kinase 1)[48], are involved in triggering a mitogen-activated protein kinase signaling cascade, clustered in a separate clade (Supplementary Fig. 7b). These findings suggest that ZmBLK1 and ZmBLK1-1 may be involved in ROS-mediated defense responses.

In SLC assays, cLUC-ZmBLK1, but not cLUC-ZmBLK1-1, interacted with ZmWIK-nLUC and ZmWAKL[Y]-nLUC, and to a lesser extent with ZmWAKL[Q]-nLUC (Fig. 4a and Extended Data Fig. 8a). Using Co-IP assays, we observed that ZmBLK1-Myc was associated with ZmWIK-GFP (Fig. 4b), and specifically with ZmWIK[ICD]-GFP, but not ZmWIK[ECD,TM]-GFP

(Fig. 4c). ZmBLK1-Myc was also associated with both ZmWAKL-GFP isoforms in a Co-IP assay (Fig. 4d). ZmBLK1 was reported to contribute to disease resistance and tethered to the plasma membrane through a post-translational N-terminal myristoylation motif[49]. This plasma membrane localization of ZmBLK1 would facilitate its interaction with ZmWIK and ZmWAKL. With all three proteins present, we detected both ZmBLK1-Myc and cLUC-ZmWIK in the complexes immunoprecipitated by either ZmWAKL[Y]-GFP or ZmWAKL[Q]-GFP in Co-IP assays (Fig. 4e). Thus, ZmBLK1, ZmWAKL and ZmWIK may form an immune signaling complex at the plasma membrane.

## ZmBLK1 is phosphorylated by ZmWAKL/ZmWIK

ZmBLK1-GST (GST, glutathione S-transferase) exhibited autophosphorylation and transphosphorylation of MBP[a], but the kinase-inactive variant ZmBLK1[K114E]-GST did not (Fig. 4f). His-ZmWIK[ICD] had self-phosphorylated and strongly phosphorylated ZmBLK1-GST and, to a lesser extent, ZmBLK1[K114E]-GST. In contrast, ZmBLK1-GST, although

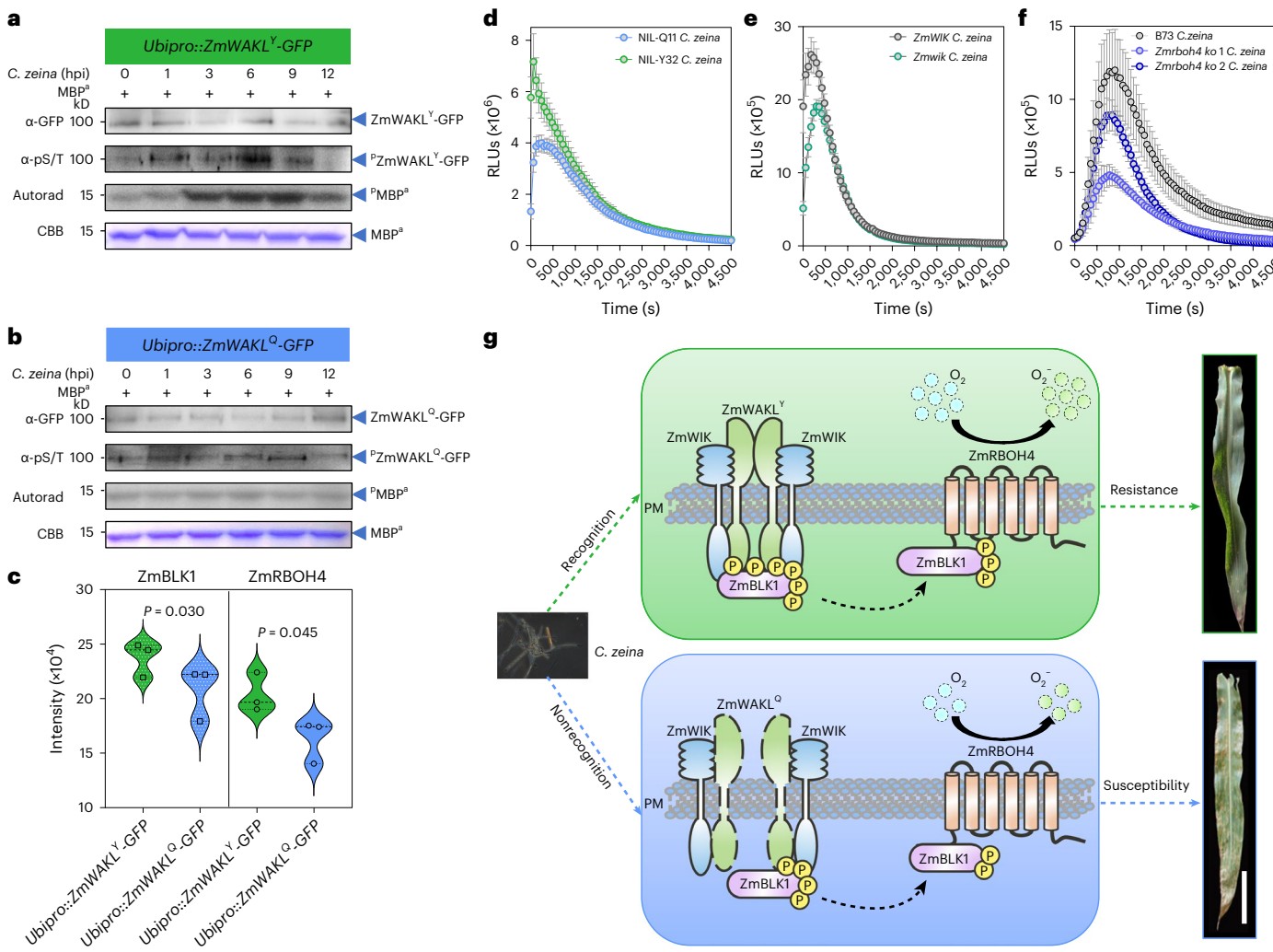

**Fig. 6 | The ZmWAKL-triggered immune response to *C. zeina* and proposed working model. a,b,** Phosphorylation and kinase activity of ZmWAKL in plants overexpressing *ZmWAKL^Y-GFP* (**a**) or *ZmWAKL^Q-GFP* (**b**) after *C. zeina* infection. The phosphorylated ZmWAKL was detected with an α-phospho-(Ser/Thr) antibody; phosphorylation was detected by autoradiography (autorad) and protein loading is shown by CBB staining. These assays were repeated two times with the same results. **c,** Protein phosphorylation intensity of ZmBLK1 and ZmRBOH4 at 6 h after *C. zeina* inoculation in phosphoproteomic analyses (*n* = 3). **d–f,** *C. zeina*-induced ROS burst in NILs (**d**, *n* = 16), *ZmWIK* mutants (**e**, *n* = 16) and

*ZmRBOH4* knockout plants (**f**, *n* = 21). The isolated leaves (V4 stage) were used to examine *C. zeina*-induced ROS, recorded as relative luminescence units (RLUs). **g,** Working model for ZmWAKL-mediated GLS resistance. PM, plasma membrane. Scale bar, 15 cm. In **c**, the violin plots show the protein phosphorylation intensity of ZmBLK1 and ZmRBOH4. The thin dashed lines indicate the interquartile range. The thick dashed line shows the medians. Statistical significance was determined by a paired *t* test. In **d–f**, data are means ± s.e. These assays were repeated three times with similar results.

showing significant autophosphorylation, hardly phosphorylated His-ZmWIK^ICD,K339E (Fig. 4g). With respect to reciprocal phosphorylation between ZmBLK1 and two ZmWAKL^ICD isoforms, ZmWAKL^Y/ICD-MBP phosphorylated ZmBLK1^K114E-GST, whereas ZmWAKL^Q/ICD-MBP didn't (Fig. 4h). In turn, ZmBLK1-GST phosphorylated ZmWAKL^Y/ICD,K391E-MBP moderately and ZmWAKL^Q/ICD,K430E-MBP more weakly (Fig. 4i). Collectively, these data suggest that reciprocal phosphorylation between the three proteins is not equivalent, with the phosphorelay being primarily directed from ZmWAKL^Y/ZmWIK to ZmBLK1.

The autophosphorylation and transphosphorylation relationships of ZmWAKL, ZmWIK and ZmBLK1 are likely to be more complex when all three proteins are present. Based on the presumptive phosphorelay pathway, we investigated the phosphorylation level of ZmBLK1 by the ZmWAKL/ZmWIK receptor complex. In the presence of the two kinase-inactive variants His-ZmWIK^ICD,K339E and ZmBLK1^K114E-GST, neither autophosphorylation nor transphosphorylation could be detected for ZmWAKL^Y/ICD-MBP (Fig. 4j(I)). The phosphorylation of

the kinase-inactive variant ZmBLK1^K114E-GST by His-ZmWIK^ICD did not change appreciably upon the addition of ZmWAKL^Y/ICD-MBP, ZmWAKL^Q/ICD-MBP or two kinase-inactive ZmWAKL^ICD-MBP variants (Fig. 4j(II)). In the presence of the kinase-inactive variant His-ZmWIK^ICD,K339E, the addition of ZmWAKL^Y/ICD-MBP, ZmWAKL^Q/ICD-MBP or two kinase-inactive ZmWAKL^ICD-MBP variants did not appear to alter the phosphorylation level of ZmBLK1-GST in any way (Fig. 4j(III)). Notably, His-ZmWIK^ICD could strongly phosphorylate ZmBLK1-GST, and this transphosphorylation was further increased upon the addition of ZmWAKL^Y/ICD-MBP, but not by the addition of ZmWAKL^Q/ICD-MBP or two kinase-inactive ZmWAKL^ICD-MBP variants (Fig. 4j(IV)).

In *N. benthamiana*, we found ZmWAKL^Y-GFP caused more phosphorylation of ZmBLK1-Myc than ZmWAKL^Q-GFP when co-expressed with cLUC-ZmWIK (Extended Data Fig. 8b). The *cLUC-ZmWIK* and *ZmBLK1-Myc* constructs were cotransfected into protoplasts isolated from transgenic plants overexpressing *ZmWAKL^Y-GFP* or *ZmWAKL^Q-GFP*. In immunoprecipitates obtained using α-Myc magnetic

beads, ZmBLK1-Myc was more phosphorylated in the presence of ZmWAKL$^Y$-GFP compared to ZmWAKL$^Q$-GFP (Extended Data Fig. 8c). The more ZmBLK1-Myc phosphorylated, the higher its kinase activity (Extended Data Fig. 8d).

We speculate that in the ZmWAKL/ZmWIK/ZmBLK1 complex, ZmWIK directly phosphorylates ZmBLK1, with ZmWAKL$^Y$ enhancing this, while ZmWAKL$^Q$ does not. These findings support an immune phosphorelay from ZmWAKL$^Y$ to ZmWIK to ZmBLK1 in GLS resistance.

## ZmBLK1 interacts with and phosphorylates ZmRBOH4

In Arabidopsis, BIK1 directly binds to and phosphorylates the N-terminal domain of RBOHD, leading to a ROS burst that mediates plant immunity[43,50]. In maize genome, six RBOHs were identified, of which ZmRBOH4 was the most closely related to AtRBOHD with 63.3% sequence identity (Extended Data Fig. 9a). In SLC assays, ZmRBOH4 interacted with ZmBLK1 but not ZmBLK1-1; furthermore, the N-terminal domain of ZmRBOH4 directly interacted with ZmBLK1, but not with two ZmWAKL isoforms or ZmWIK (Fig. 5a and Extended Data Fig. 9b,c). We confirmed these results by pull-down assays (Fig. 5b and Extended Data Fig. 9d). For the other five ZmRBOHs, ZmRBOH1/ZmRBOH5/ZmRBOH6 but not ZmRBOH2/ZmRBOH3 interacted with ZmBLK1 in SLC assays (Extended Data Fig. 9e). However, unlike *ZmRBOH4*, which has high and pathogen-inducible gene expression in leaves (Extended Data Fig. 9f), *ZmRBOH1* showed relatively low gene expression, which could not be induced by pathogen inoculation, and ZmRBOH5/ZmRBOH6 were exclusively expressed in the anthers (Supplementary Fig. 8a,b). Hence, only ZmRBOH4 is likely involved in the immune signaling against GLS.

Both ZmWAKL$^{Y/ICD}$-MBP and His-ZmWIK$^{ICD}$ showed autophosphorylation, but phosphorylated His-N-ZmRBOH4 not at all or weakly, respectively (Fig. 5c,d). This is consistent with the lack of direct interaction between ZmRBOH4 and ZmWAKL$^Y$ (or ZmWIK; Extended Data Fig. 9b–d). By contrast, ZmBLK1-GST strongly phosphorylated His-N-ZmRBOH4 (Fig. 5e). When all four proteins, ZmWAKL, ZmWIK, ZmBLK1 and ZmRBOH4, were mixed in vitro, their phosphorylation levels varied substantially, from strongly phosphorylated ZmBLK1-GST and His-N-ZmRBOH4 to barely phosphorylated His-ZmWIK$^{ICD}$ to undetectably phosphorylated ZmWAKL$^{Y/ICD}$-MBP. Phosphorylation of His-N-ZmRBOH4 was somewhat enhanced in the presence of ZmWAKL$^{Y/ICD}$-MBP compared to ZmWAKL$^{Q/ICD}$-MBP and the two kinase-inactive ZmWAKL$^{ICD}$-MBP variants (Fig. 5f). In the presence of kinase-active ZmWAKL$^{Y/ICD}$-MBP and ZmBLK1-GST, His-N-ZmRBOH4 displayed weaker phosphorylation when His-ZmWIK$^{ICD}$ was replaced by its kinase-inactive variant His-ZmWIK$^{ICD,K339E}$. However, the substitution of ZmBLK1-GST by its kinase-inactive variant ZmBLK1$^{K114E}$-GST almost eliminated His-N-ZmRBOH4 phosphorylation. When both kinase-inactive His-ZmWIK$^{ICD,K339E}$ and ZmBLK1$^{K114E}$-GST variants were present, no His-N-ZmRBOH4 phosphorylation could be detected (Fig. 5g). We interpreted these results as evidence that ZmBLK1 is the major upstream kinase phosphorylating N-ZmRBOH4-GST in this assay, and ZmRBOH4 may join the ZmWAKL$^Y$/ZmWIK/ZmBLK1 complex to form an immune module.

To verify the function of *ZmRBOH4*, we knocked out *ZmRBOH4* in B73 using CRISPR/Cas9. Notably, these two knocked out lines exhibited significantly reduced plant height and increased GLS susceptibility compared with wild-type plants under artificial inoculation (Fig. 5h,i and Extended Data Fig. 10), indicating ZmRBOH4 is involved in regulating GLS resistance.

## ZmWAKL-mediated immunity against GLS in vivo

As noted above, *ZmWAKL* could be induced by *C. zeina* infection with a peak at 6 hpi (Extended Data Fig. 6b). Here we wanted to know what would happen to ZmWAKL$^Y$ and ZmWAKL$^Q$ proteins in response to *C. zeina* infection. Transgenic plants overexpressing *ZmWAKL$^Y$-GFP* inoculated with *C. zeina* showed increased levels of phosphorylated

ZmWAKL$^Y$, peaking at 6 hpi, with stronger kinase activity (Fig. 6a). However, in a parallel experiment, ZmWAKL$^Q$-GFP appeared to be unaffected by *C. zeina* infection, exhibiting only basal phosphorylation level and kinase activity (Fig. 6b). Like what was previously observed (Fig. 2h), we speculate that coprecipitated protein(s) in the ZmWAKL$^Q$-GFP immunoprecipitate, rather than ZmWAKL$^Q$-GFP itself, may be responsible for this basal activity. To gain insight into the phosphorylation changes of the other proteins, we performed proteomic and phosphoproteomic analyses on transgenic plants overexpressing *ZmWAKL$^Y$-GFP* and *ZmWAKL$^Q$-GFP* collected at 6 hpi. We detected that the phosphorylation levels of ZmBLK1 and ZmRBOH4 showed significant increases in *ZmWAKL$^Y$-GFP* plants compared to those in *ZmWAKL$^Q$-GFP* plants (Fig. 6c and Supplementary Table 5).

Considering our evidence that immune signaling, initiated by ZmWAKL$^Y$ in response to *C. zeina* attack, eventually converges to ZmRBOH4 via ZmWIK and ZmBLK1, it appears that GLS resistance may be associated with a ROS burst. We observed that NIL-Y32 exhibited higher *C. zeina*-induced ROS levels compared to NIL-Q11 (Fig. 6d) and *Zmwik* and *Zmrboh4* null mutants showed lower *C. zeina*-induced ROS levels than wild-type plants (Fig. 6e,f).

## Working model of maize WAKL–WIK–BLK1–RBOH4 immune module

We characterized the role of the cell-wall-associated RLK ZmWAKL in QDR to GLS in maize. Our investigations suggest the following model: ZmWAKL$^Y$ could self-associate and interact with the coreceptor ZmWIK to form the ZmWAKL$^Y$/ZmWIK receptor complex at the plasma membrane. Upon challenge by the pathogen *C. zeina*, ZmWAKL$^Y$ is activated and transiently increases its phosphorylation activity, and together with ZmWIK, transmits immune signals to ZmBLK1 and then to ZmRBOH4, which triggers a ROS burst to incur innate immunity (Fig. 6g). In contrast to ZmWAKL$^Y$, ZmWAKL$^Q$ is unable to associate with itself, shows no autophosphorylation or kinase activity and interacts weakly with ZmWIK. Upon pathogen challenge, ZmWAKL$^Q$ shows no detectable increase in its phosphorylation activity, thus failing to trigger immune responses, leading to GLS susceptibility (Fig. 6g). The maize ZmWAKL–ZmWIK–ZmBLK1–ZmRBOH4 immune module perceives pathogen invasion, transduces immune signals and activates defense responses, eventually conferring quantitative resistance to GLS.

## Discussion

Identifying the causative gene underlying QDR is challenging due to the difficulties with the genetic dissection and verification of this quantitative trait. Here we adopted a sequential fine-mapping strategy and multiple transgenic approaches to demonstrate that *ZmWAKL* is the causative gene for *qRgls1* (Fig. 1). Characterizing ZmWAKL allowed us to identify its coreceptor ZmWIK, a plasma-membrane-localized RLK (Figs. 2 and 3). Thereafter, partly based on analogies to systems found in Arabidopsis, we identified the ZmBLK1 and ZmRBOH4 as downstream components of the ZmWAKL$^Y$/ZmWIK receptor complex (Figs. 4 and 5).

Extracellular ROS are produced primarily through activation of RBOHs at the plasma membrane and act as a pivotal early component of PTI responses[51]. Besides, RBOHs are crucial for growth and development[51,52], like knocking out *ZmRBOH4* results in weaker plants (Extended Data Fig. 10). The reduced vitality of ZmRBOH4 mutants might also contribute to the increased GLS susceptibility (Fig. 5h,i).

In cereal crops, the molecular mechanism of ROS production and its contribution to QDR are not completely understood. Therefore, the discovery of the maize ZmWAKL–ZmWIK–ZmBLK1–ZmRBOH4 immune module will have a significant influence on the conceptual underpinnings of QDR in plants.

## Online content

Any methods, additional references, Nature Portfolio reporting summaries, source data, extended data, supplementary information,

acknowledgements, peer review information; details of author contributions and competing interests; and statements of data and code availability are available at https://doi.org/10.1038/s41588-023-01644-z.

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

## Methods

### Plant materials

In our previous study, the major QTL-*qRgls1* is restricted to a ~1.4-Mb region flanked by makers GZ204 and IDP5 (ref. [22]). To fine-map *qRgls1*, the recombinants were self-pollinated and backcrossed to Q11 to develop $BC_2F_5$ (1,865 plants), $BC_5F_7$ (1,055 plants) and $BC_7F_7$ (1,162 plants) populations. The five $BC_7F_7$ recombinants were subsequently backcrossed to Q11 to generate $BC_8F_7$ generations, which were then self-pollinated to develop $BC_8F_8$ homozygous lines with/without Y32 segments. Comparison of Y32 donor regions with GLS resistance allowed us to confirm the final *qRgls1* region. From the recombinant IV, we obtained a pair of near-isogenic lines differing at *qRgls1*, namely NIL-Y32 (with *qRgls1*) and NIL-Q11 (without *qRgls1*). The two NILs had 96.12% genetic identity (as assessed with the Axiom_M6H60k genotyping chip, Thermo Fisher Scientific) and were used for exploring the molecular mechanism underlying *ZmWAKL*-mediated innate immunity. In addition, a collection of 98 maize inbred lines (Supplementary Table 2) was used for examining allelic variations of the *ZmWAKL* locus.

The EMS mutant line *Zmwik* was obtained from the Maize EMS-induced Mutant Database (http://www.elabcaas.cn/memd/public/index.html#/; mutant ID: EMS4-0073f9). The mutant was generated in the B73 background and contained the heterozygous *ZmWIK* locus. The mutant was backcrossed twice to NIL-Y32, followed by self-pollination, to introduce *ZmWAKL*[Y] into the *ZmWIK* wild-type and its homozygous *Zmwik* mutant to evaluate GLS resistance.

### Field assays

The fine-mapping populations, maize inbred lines and *ZmWIK* mutants were planted in Baoshan (Yunnan, China) and naturally infected with the pathogenic fungus *C. zeina*. Baoshan is the ideal place for GLS development due to sufficient natural disease stress. The seeds were sown at 5 m in length with 0.5 m between rows and 0.25 m between plants within a row.

In Beijing, transgenic plants and NILs were planted in the same way as in Baoshan, except for using the 4-m long plot and being artificially inoculated with *C. zeina*. The GLS inoculum was prepared using the pathogenic fungus *C. zeina*. The fungus was cultured on potato dextrose agar medium, and the spores were washed off with $ddH_2O$ to prepare a spore inoculum at a concentration of $5 \times 10^3$ spores per ml. Plants were inoculated at the 10-leaf stage by pouring 5-ml inoculum into the leaf whorl. Samples were collected at different time points after inoculation, as specified in each experiment. Each sample contains leaf tissues collected from at least three individual plants. For gene expression analysis, one leaf punch per plant was collected from the middle of the inoculated leaf, and each genotype/treatment sample contained three punches, which were pooled and stored at −80 °C. For scanning electron microscope (SEM) analysis, four punches across the center of each leaf were collected.

The GLS symptoms are scored three times at 1-week intervals, starting 2 weeks after pollination. The number and size of lesions on the leaves of the entire plant were used to assess the GLS severity. GLS scale was rated using the following scale: 1 (highly resistant), 3 (resistant), 5 (intermediate resistant/susceptible), 7 (susceptible) and 9 (highly susceptible). To convert the scale to a DSI, scales 1, 3, 5, 7 to 9 were assigned to a distinct numerical value of 0, 0.25, 0.5, 0.75 and 1, respectively.

$$DSI(\%) = \frac{\sum \text{Assigned disease severity value} \times \text{number of plants in corresponding grade}}{1 \times \text{total number of plants}} \times 100$$

Days to heading, silking and anthesis were recorded for each plant from NILs and homozygous transgenic lines.

### Sequential fine-mapping of *qRgls1*

Newly developed markers in the *qRgls1* region were used to genotype recombinants (Supplementary Table 1). In each generation, new recombinants within the mapped *qRgls1* region were selected and backcrossed to Q11 to produce progeny for fine-mapping. The *qRgls1* locus was fine-mapped by comparing the genotypes of the recombinants with their deduced phenotypes based on the resistant performance of their progeny. A significant difference in the DSI between the heterozygous and homozygous genotypes of the offspring indicates the presence of the resistance allele at *qRgls1* in the heterozygous region of their parental recombinant; otherwise, there is no resistance allele at *qRgls1*.

### Genetic transformation

From the positive Y32 BAC clone 57-9-1-93, a 9.2 kb genomic fragment, containing the 2.1-kb promoter, 2.2-kb *ZmWAKL*[Y] coding and 4.9-kb 3′-UTRs, was subcloned into pCAMBIA3301 according to the method described previously[8]. To construct overexpression vector, the fused gene of *ZmWAKL* coding sequence (CDS) and EGFP was cloned into the express vector pBCXUN-Myc driven by the maize ubiquitin promoter as described earlier[53]. In addition, we constructed overexpression vectors for the chimeric genes *ZmWAKL*[C], *ZmPR5L* and *ZmWIK*. To knockout the endogenous genes *ZmPR5L* and *ZmRBOH4*, CRISPR/Cas9 vectors were constructed following the protocol described previously[54]. These vectors were transferred to *Agrobacterium* strain EHA105 and then transformed into the maize inbred line B73.

The genotypes were confirmed by PCR or sequencing standard PCR products. All $T_1$-positive or edited plants from each transgenic event were self-pollinated to produce homozygous transgenic plants or backcrossed to the susceptible parent Q11 or NIL-Y32 to produce backcross populations for functional verification.

### Plant protein extraction and immunoblot analysis

Plant total protein was extracted from maize or tobacco leaves. The tissue sample was ground in liquid nitrogen and resuspended in an equal volume (1:1 fresh wt/vol) of the extraction buffer (50 mM Tris−HCl (pH 7.5), 150 mM NaCl, 1 mM dithiothreitol (DTT), 1 mM phenylmethylsulfonyl fluoride (PMSF) and 0.2 % Triton X-100) on ice for at least 30 min and then centrifuged at 12,000$g$ for 30 min at 4 °C to obtain the supernatant. The remaining supernatant was incubated with ~5 μl magnetic beads for 2 h at 4 °C. The beads were magnetically separated, washed twice and then heated in 100 μl of 1× SDS sample buffer at 95 °C for 5 min. The protein sample was subjected to electrophoresis on a 10% (wt/vol) polyacrylamide gel and transferred to a polyvinylidene fluoride blotting membrane (GE HealthCare, A10116498). The membrane was incubated for 1 h at room temperature with 5% nonfat milk in tris-buffered saline (20 mM Tris−HCl (pH 7.6) and 150 mM NaCl) containing 0.05% Tween 20 (TBST). All antibody incubations were performed in 1× phosphate-buffered saline (1× PBS; 137 mM NaCl, 2.7 mM KCl, 4.3 mM $Na_2HPO_4$, 1.4 mM $KH_2PO_4$, pH 7.4). The fused protein was detected with the corresponding tag antibody at a dilution of 1:2,000 (vol/vol) overnight at 4 °C. α-Actin (dilution of 1:2,000 (vol/vol)) was used as an endogenous control (EASYBIO, BE0028). The membrane was incubated for 1 h with an immunoglobulin horseradish peroxidase-conjugated secondary antibody at a dilution of 1:5,000 (vol/vol) to visualize the signal.

### Subcellular localization

The CDS of *ZmWAKL*[Y], *ZmWAKL*[Q] and ZmWIK were inserted into the vector pEZS-NL to create EGFP-fused expression constructs driven by the 35S promoter. Similarly, the *ZmWIK* CDS was cloned into pSuper1300-GFP to generate the fusion construct. The *ZmWIK-GFP* constructs were transformed into tobacco leaves using *Agrobacterium*-mediated method. The *ZmWAKL*[Y]-*EGFP*, *ZmWAKL*[Q]-*EGFP* and *ZmWIK-EGFP* constructs were also mixed with nano-grade gold

particles and bombarded into onion epidermal cells, which were then cultured at 28 °C in the dark for 8–10 h. The GFP signals were visualized under a confocal microscope (Zeiss). Plasmolysis was achieved by incubating onion epidermal cells in 0.3 g ml$^{-1}$ sucrose for 5 min.

Fluorescence was examined in the root epidermal tissues of 1-week-old *ZmWAKL$^Y$-GFP* or *ZmWAKL$^Q$-GFP* overexpression transgenic plants and B73 under a confocal microscope. The membrane-impermeable dye FM4-64 (MedchemExpress, HY-103466) was used as the plant membrane marker.

#### Phylogenetic analysis
The amino acid sequences of reported immune-related WAKs/WAKLs and RLCKs were downloaded from the National Center for Biotechnology Information (NCBI) database (https://www.ncbi.nlm.nih.gov/). The ZmRBOH1-6 amino acid sequences were retrieved from the Gramene database (http://www.gramene.org/). The full-length sequences were used for the phylogenetic analysis, and the phylogenetic trees were constructed using the neighbor-joining method in MEGA 7.0 (http://www.megasoftware.net). Bootstrap values from 1,000 pseudo-replicates were used to provide support for the nodes in the phylogenetic tree. The conserved domains were predicted using the NCBI Conserved Domains Database (https://www.ncbi.nlm.nih.gov/Structure/cdd/wrpsb.cgi).

#### Yeast two-hybrid assay
The DUAL membrane system was used to verify the self-association of ZmWAKL. The signal peptide sequences of *ZmWAKL$^Y$* and *ZmWAKL$^Q$* were removed, and the remaining sequences were cloned into the pBT3-SUC vector. The full-length CDS of *ZmWAKL$^Y$* and *ZmWAKL$^Q$* were cloned into the pPR3-N vector. These vectors were cotransformed into the yeast strain NMY51 to detect protein–protein interactions according to the manufacturer's user guide (Clontech). Then, the GUB domain (amino acids 46–155 in ZmWAKL$^Y$ and amino acids 37–182 in ZmWAKL$^Q$) and the intracellular kinase domain (amino acids 362–630 in ZmWAKL$^Y$ and amino acids 404–666 in ZmWAKL$^Q$) were cloned into pGADT7 and pGBKT7 vectors, respectively, to verify the interaction domains. These vectors were cotransformed into the Y2HGold strain, and the protein interactions were tested on a selective medium.

#### SLC assay
The CDS of *ZmWAKL$^Y$*, *ZmWAKL$^Q$*, *ZmBLK1*, *ZmWIK*, *ZmRBOH4* and their gene segments were cloned into the JW771-35S-CLuc vector (cLUC) and JW772-35S-NLuc vector (nLUC), respectively, to generate fusion proteins with the C-terminal or N-terminal fragment of the luciferase gene. ZmWIK and ZmWAKL were divided into their ECD/TM and ICD, marked as ZmWIK$^{ECD,TM}$ (1-269 aa) and ZmWIK$^{ICD}$ (270-594 aa), ZmWAKL$^{Y/ECD,TM}$ (1-324 aa) and ZmWAKL$^{Y/ICD}$ (325-665 aa), ZmWAKL$^{Q/ECD,TM}$ (1-363 aa) and ZmWAKL$^{Q/ICD}$ (364-704 aa). The N-terminal domain was marked as N-ZmRBOH4 (1-377 aa). The pairs of constructs were co-infiltrated into *N. benthamiana* leaves using the previously reported method[55]. Two days after inoculation, 1 mM luciferin (Promega, E1601) was sprayed onto the inoculated leaves, and the luminescence signal was measured using the Chemiluminescent Imaging System (Tanon).

The luminescence signals were analyzed using ImageJ Launcher software (National Institutes of Health) to determine their intensities, so-called the mean gray value. This assay was immediately followed by protein extraction and immunoblotting with α-LUC antibody (Abcam, ab181640), and the immunoblot bands were measured using ImageJ Launcher as well. Total protein of the infiltrated leaf tissues was extracted and detected as described above.

#### Co-IP assay
To generate *ZmWAKL$^Y$-Myc*, *ZmWAKL$^Y$-GFP*, *ZmWAKL$^Q$-Myc*, *ZmWAKL$^Q$-GFP*, *ZmWIK-Myc*, *ZmWIK$^{ECD/TM}$-GFP*, *ZmWIK$^{ICD}$-GFP* and

*ZmBLK1-Myc*, the CDS or gene segments for *ZmWAKL*, *ZmWIK* and *ZmBLK1* were amplified and then cloned into *pSuper1300* (Myc-tag or GFP-tag). The resulting plasmids were transformed into *N. benthamiana* leaves and expressed for about 48 h. Total protein from the infiltrated leaf tissues was extracted as described above, and the supernatant was incubated with the α-GFP magnetic beads (MBL, D153-11) at 4 °C for 2 h. The products were analyzed and detected by immunoblotting with α-Myc antibody (ABclonal, AE010) and α-LUC antibody (Abcam, ab181640). GFP and GFP-fusion proteins were detected by immunoblotting with α-GFP antibody (ABclonal, AE012).

#### IP–MS
The immunocomplexes were analyzed by mass spectrometry (MS) at the China Agricultural University Mass Spectrum Laboratory. Briefly, ~2 g leaf of *Ubipro:ZmWAKL$^Y$-GFP* transgenic plants were collected and ground in a mortar using liquid nitrogen. The GFP-tagged fusion protein complex was extracted as described above and separated by sodium dodecyl-sulfate polyacrylamide gel electrophoresis (SDS–PAGE). The gel lanes were cut, sliced and subjected to in-gel digestion with trypsin. After destaining and tryptic digestion, peptides were extracted and redissolved in 25 μl (0.1%) trifluoroacetic acid. In total, 6 μl of extracted peptides were analyzed by LTQ Orbitrap Velos mass spectrometer (Thermo Fisher Scientific). The Mascot search engine Mascot Server 2.3 (Matrix Science) was used for protein identification by searching against the UniProt protein database (https://www.uniprot.org/). The false discovery rate (FDR) was also set to 0.01 for protein identifications. The significance threshold was set at $P < 0.05$, and a minimum number of significant unique sequences was set to 1.

#### In vitro pull-down assay
In vitro pull-down assay was carried out as previously described[56]. In brief, 0.5 μg of His-ZmWAKL$^{Y/ICD}$ or His-ZmWAKL$^{Q/ICD}$ or His-TF-ZmWAKL$^{Y/ICD}$ or His-TF-ZmWAKL$^{Q/ICD}$ or His-ZmWIK$^{ICD}$ or His-ZmBLK1 protein was incubated with 5 μg of ZmWIK$^{ICD}$-MBP or N-ZmRBOH4-GST or N-ZmRBOH4-MBP and immunoprecipitated by MBP or GST agarose resin at 4 °C for 2 h. The mixture was gathered by centrifugation at 500$g$ for 5 min, followed by washing with phosphate-buffered saline (PBS) buffer five times. Proteins were separated by 10% (wt/vol) SDS–PAGE and detected with α-MBP antibody (ABclonal, AE016), α-GST antibody (Yeasen, 30903ES10) and α-His antibody (Yeasen, 30404ES60).

#### In vitro phosphorylation assay
ZmWAKL$^{Y/ICD}$-MBP, ZmWAKL$^{Q/ICD}$-MBP, His-ZmWIK$^{ICD}$, ZmBLK1-GST and their corresponding kinase-inactive variants, His-N-ZmRBOH4 fusion protein and MBP$^a$ (EMB Millipore, 13-104) were used for in vitro phosphorylation assay in the presence of ($^{32}$P)-γ-ATP.

For the in vitro phosphorylation assay, the recombinant proteins were incubated in a kinase buffer (20 mM Tris–HCl (pH 7.5), 10 mM MgCl$_2$, 1 mM CaCl$_2$ and 1 mM DTT) in the presence of 1 mM unlabeled ATP and 1 μCi of ($^{32}$P)-γ-ATP for 30 min at 30 °C. The reactions were stopped by adding 5× SDS loading buffer (GenStar, E153-10). Proteins were separated by SDS–PAGE, followed by staining with CBB overnight with decolorization. The phosphorylation status of the fusion proteins was analyzed by autoradiography using a Typhoon 9410 Variable Mode Imager (GE HealthCare).

#### In vivo phosphorylation assay
ZmBLK1 protein phosphorylated in vivo was detected using α-phospho-(Ser/Thr) antibody (Abcam, ab117253). Briefly, proteins were extracted from the protoplasts of *Ubipro:ZmWAKL$^Y$-GFP* and *Ubipro:ZmWAKL$^Q$-GFP* transgenic plants co-expressing *cLUC-ZmWIK* and *ZmBLK1-Myc*. Proteins were also extracted from the *N. benthamiana* leaves co-expressing *cLUC-ZmWIK*, *ZmBLK1-Myc* and *ZmWAKL-GFP* or *GFP*. The ZmWAKL protein phosphorylation level

was identified by western blotting with α-phospho-(Ser/Thr) antibody (Abcam, ab117253). The proteins were extracted from the samples collected at different times after *C. zeina* infection. Protein extraction and detection were consistent with the methods described above.

## Sample preparation for phosphoproteomic and proteomic

The sample was first grinded with liquid nitrogen, then the powder was added to four volumes of lysis buffer (including 10 mM dithiothreitol, and 1% protease inhibitor cocktail, 50 µM PR-619, 3 µM tryptic soy agar (TSA), 50 mM N-acetylmuramic acid (NAM) and 1% phosphatase inhibitor), ultrasonic cracking. An equal volume of Tris-saturated phenol (pH 8.0) was added and centrifuged at 5,500g at 4 °C for 10 min. Then the supernatant was collected and five times the volume of 0.1 M ammonium acetate/methanol was added to precipitate overnight. The precipitates were washed with methanol and acetone, respectively. Finally, the precipitate was redissolved in 8 M urea, and the protein concentration was determined with bicinchoninic acid (BCA) kit according to the manufacturer's instructions. All samples were labeled with TMT 6-plex reagent (Thermo Fisher Scientific, 90068) according to the manufacturer's instructions. The labeled samples were combined and dried for further analysis.

## Identification of peptides and phosphopeptides

The mixed TMT 6-labeled samples were analyzed by high pH reverse phase high-performance liquid chromatography (HPLC) with Agilent 300 Extend C18 as the chromatographic column. The peptide segments were separated using a gradient ranging from 8% to 32% acetonitrile (pH 9.6) within 60 min. The separated peptide segments were combined into eight components. The combined components were subjected to vacuum freeze-drying, and subsequent operations were carried out. Peptide mixtures were first incubated with IMAC microspheres suspension with vibration in loading buffer (50% acetonitrile/0.5% acetic acid). To remove the nonspecifically adsorbed peptides, the immobilised metal affinity chromatography (IMAC) microspheres were washed with 50% acetonitrile/0.5% acetic acid and 30% acetonitrile/0.1% trifluoroacetic acid, sequentially. To elute the enriched phosphopeptides, an elution buffer containing 10% $NH_4OH$ was added and the enriched phosphopeptides were eluted with vibration. The supernatant containing phosphopeptides was collected and lyophilized for liquid chromatography with tandem mass spectrometry (LC–MS/MS) analysis. All raw data were retrieved using Proteome Discoverer (v2.4.1.15). Database is *Zea_mays*_4577_PR_20221123 (6,3235 sequences). A reverse library was added to the database to calculate the false positive rate (FPR) caused by random matching, and common contaminated libraries were added to the database to eliminate the impact of contaminated proteins in identification results. The quantitative method is set to TMT 6-plex, and the FPR for protein, peptide and peptide-spectrum match (PSM) identification is set to 1%.

## Oxidative burst assay

To determine pathogen-induced ROS accumulation, the GLS pathogen *C. zeina* was used to activate the innate immune system. *C. zeina* was cultured on the corn leaf powder medium for 1–2 weeks. Then, everything on the surface of the medium, including spores and hyphae, was eluted with sterile water, and the eluted solution's $OD_{600}$ was adjusted to 2. The oxidative burst assay was carried out as previously described with slight modifications[21]. The fourth fully expanded leaves were sliced into 3 mm strips and incubated in 200 µl 1% DMSO overnight. In total, 5 µl *C. zeina* was added and treated for 2 h. Then 1% DMSO was replaced by 100 µl 2× L-012 (Wako, 120-04891) in 0.05% Silwet L-77, and the strip samples were infiltrated for 1 h. Then 100 µl buffer containing 40 µg ml$^{-1}$ horseradish peroxidase (Sigma-Aldrich, V900503) was added into the reactions. The luminescence was recorded every 60 s for 1.25 h using the GLOMAX96 Luminometer (Promega).

## Inclusion and ethics

This study's data exclusively comes from corn, without involving any animal experiments. The transgenic planting and artificial inoculation processes are subject to strict regulation. All experimental data are included in the Data availability section.

## Statistics analysis

*P* values and sample sizes (*n*) are indicated in individual figures and figure legends. Statistical analysis was performed by IBM SPSS Statistics SV26. Statistical differences between the two groups were analyzed by two-sided Student's *t* test or paired *t* test. Statistical significance between more than two groups was analyzed based on one-way analysis of variance (ANOVA) with Tukey's test or Fisher's least significant difference (LSD) test. Different lowercase letters indicate a significant difference ($P < 0.05$).

## Reporting summary

Further information on research design is available in the Nature Portfolio Reporting Summary linked to this article.

## Data availability

The authors declare that the data supporting the findings of this study are available within the paper and its Supplementary Information files. The reported WAKs/WAKLs and RLCKs' protein sequences are downloaded from the NCBI (http://www.ncbi.nlm.nih.gov/). The protein sequences of AtRBOHs are downloaded from the Arabidopsis Information Resource database (TAIR, https://www.arabidopsis.org/) and the protein sequences of ZmRBOHs are obtained from the Gramene database (https://www.gramene.org/). The expression data of ZmRBOHs and RLKs are obtained from the Plant Public RNA-seq database (http://ipf.sustech.edu.cn/pub/plantrna/). The B73 genomic sequences in the mapped *qRgls1* region are collected from the MaizeGDB (https://www.maizegdb.org/). The sequences of two BAC clones (17-37-1-53 and 57-9-1-93) are available at GenBank accessions OQ435908 and OQ435909, respectively. The genomic sequences of *ZmWAKL* used for haplotype analysis are available at GenBank under accessions OQ425304–OQ425401. The coding sequences of *ZmWAKL$^Y$* and *ZmWAKL$^Q$* are available at GenBank accessions OQ421108 and OQ421109, respectively. The genomic and coding sequences of *ZmPR5$^Y$* and *ZmPR5$^Q$* are available at GenBank accessions OQ421106–OQ421107 and OQ421110–OQ421111, respectively. The coding sequence of *ZmWIK* is available at GenBank accession OQ421112. The coding sequences of *ZmBLK1* and *ZmBLK1-1* are available at GenBank accessions OQ421113 and OQ421114, respectively. The N-terminal of ZmRBOH4 is available at GenBank accession OQ421115. Source data are provided with this paper.

## Code availability

All software used in this study is publicly available on the Internet as described in the Methods and Reporting summary.

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

## Acknowledgements

We thank the members of Xu Laboratory for their helpful discussions. We thank the Center for Crop Functional Genomics and Molecular

Breeding of China Agricultural University for providing transgenic technology support. We also thank W. Zhu (China Agricultural University) and G. Bi (China Agricultural University) for their insightful discussions during manuscript preparation. This research was supported by the National Key Research and Development Program (grant 2022YFD1201800) and the National Natural Science Foundation of China (grant 31471500).

## Author contributions

M.L.X. and T.Z. conceived the project and designed the experiments. M.L.X. supervised this project. T.Z. performed most of the experiments. M.Z. was involved in most of the fieldwork. Y.Z. and L.X. were involved in the fine-mapping process and helped revise the manuscript. Q.Q.Z. was involved in the phosphorylation assay and S.N.D. was involved in confocal microscopy experiments and protein expression. C.Y.G. identified the interaction between ZmBLK1 and ZmRBOH4. T.T.L. and Y.C.L. contributed to the materials management. X.M.F. and Y.Q.B. provided the materials and supported the phenotypic identification. P.B.-K. participated in project discussions and manuscript revision. M.L.X. and T.Z. wrote the manuscript with input from all authors.

## Competing interests

The authors declare no competing interests.

## Additional information

**Extended data** is available for this paper at https://doi.org/10.1038/s41588-023-01644-z.

**Correspondence and requests for materials** should be addressed to Mingliang Xu.

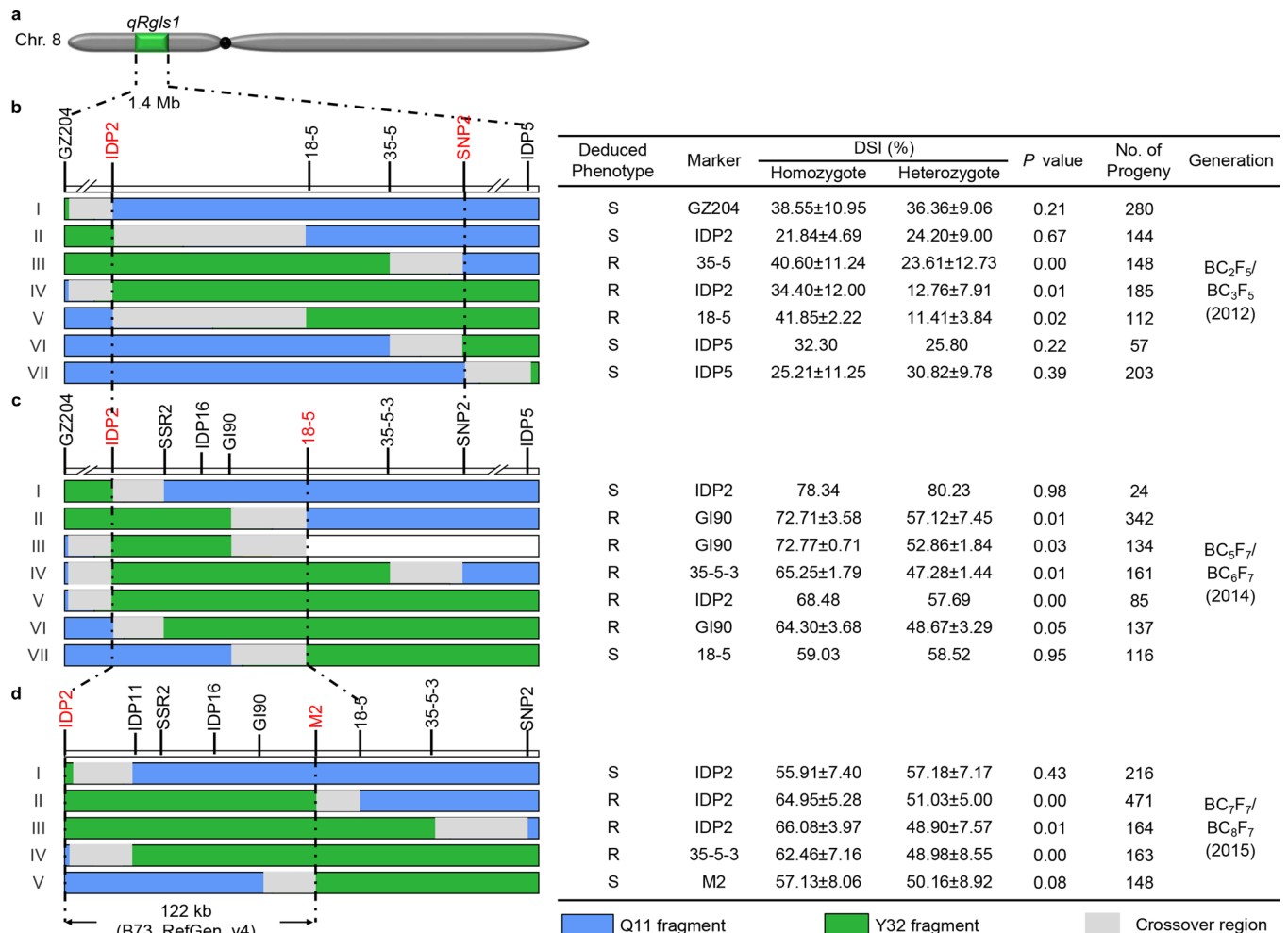

**Extended Data Fig. 1 | Sequential fine-mapping of *qRgls1*. (a)** Previous mapping efforts located *qRgls1* to a 1.4-Mb region on the short arm of maize chromosome 8. **(b–d)** Sequential fine-mapping of *qRgls1* (n = 3290). The schematic diagrams to the right represent the genomic architecture of each recombinant type. The markers used in the fine-mapping are indicated, with red markers showing the limits of the *qRgls1* region as defined at each fine-mapping step. The tables to the right summarize the DSI from homozygous and heterozygous backcross progeny of recombinants. Significant difference (*P* < 0.05) in DSI between heterozygous and homozygous offspring indicates the presence of the resistance allele at *qRgls1* in the heterozygous region of their parental recombinant; otherwise, there is no *qRgls1* (*P* > 0.05). Data are shown as means ± s.d. in (**b**)–(**d**) and the *P* value was determined based on two-sided Student's *t*-test.

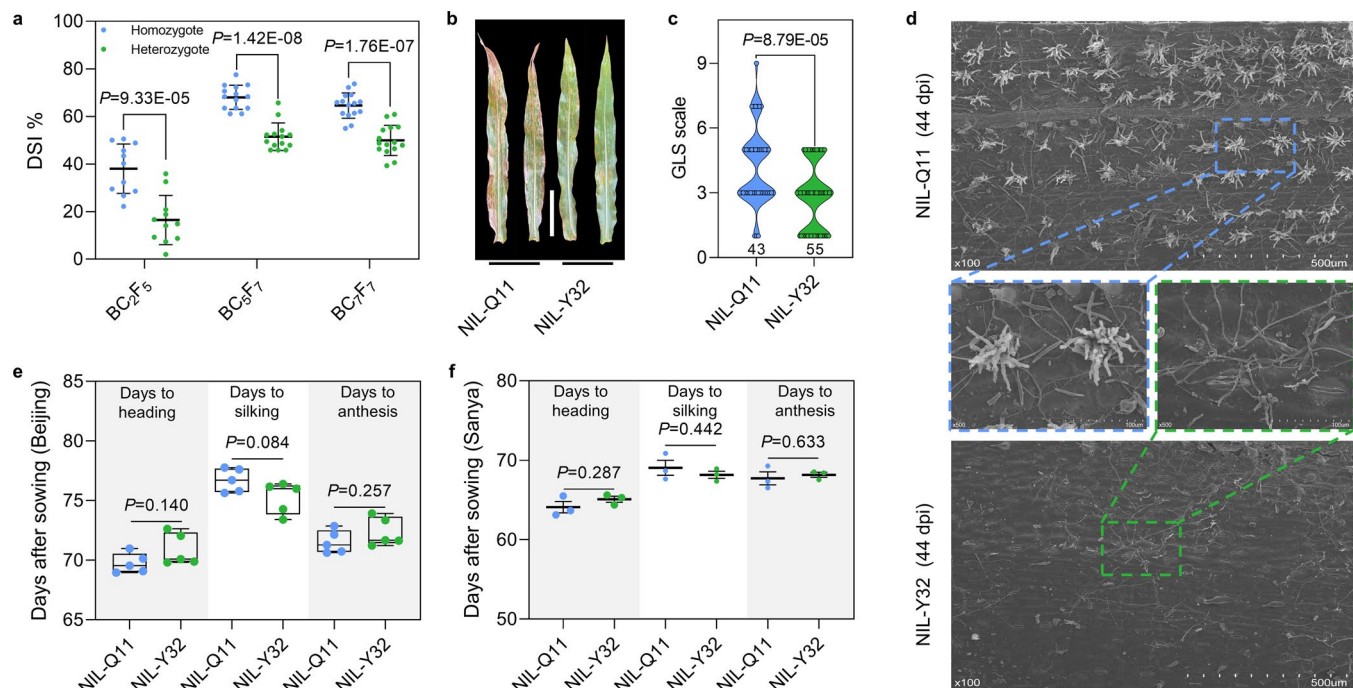

**Extended Data Fig. 2 | The genetic effect and phenotype of *qRgls1*.** (**a**) Genetic effect of *qRgls1* on maize resistance to gray leaf spot in field tests. (**b,c**) GLS symptoms (**b**) and GLS scale (**c**) (n = 98) of NIL-Q11 and NIL-Y32 plants at 44 dpi in the field. Scale bar, 15 cm. (**d**) *C. zeina* hyphae on the leaf surface of NIL-Q11 and NIL-Y32 plants at 44 dpi. Scale bars, 500 μm (overall view); 100 μm (detailed view). At least three separate leaves were observed with similar results. (**e,f**) Pairwise comparison of flowering-related traits of two NILs in Beijing (long days)

(n = 5) and Sanya (short days) (n = 3). Data are shown as means ± s.d. in (**a**), means ± s.e. in (**f**). In (**c**), data are shown as violin plots with individual data points. In (**a**), (**c**) and (**e**) the *P* value was determined based on two-sided Student's *t*-test. Data in (**e**) are displayed as box and whisker plots with individual data points. The error bars represent maximum and minimum values. Center line, median; box limits, 25th and 75th percentiles.

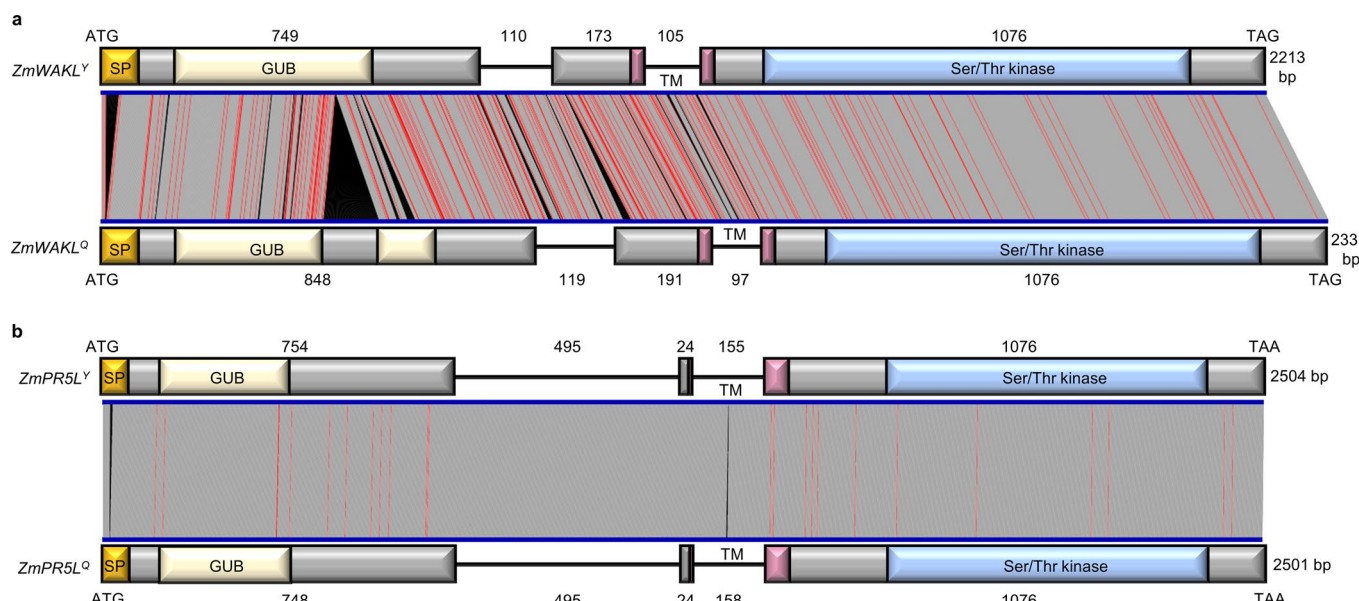

**Extended Data Fig. 3 | The schematic diagrams of *ZmWAKL* and *ZmPR5L* loci.** Schematic diagrams of *ZmWAKL* (**a**) and *ZmPR5L* (**b**) loci, as determined by aligning the full-size complementary DNA (cDNA) with the corresponding genomic sequence. Rectangles, exons; lines, introns. The colors indicate different domains: yellow, SP (signal peptide); light orange, GUB (galacturonan-binding domain of wall-associated receptor kinase); pink, TM (transmembrane domain); blue, Ser/Thr kinase. Gray shading between the Y32 and Q11 alleles represents the syntenic region; red lines denote SNPs; black spaces or lines represent InDels.

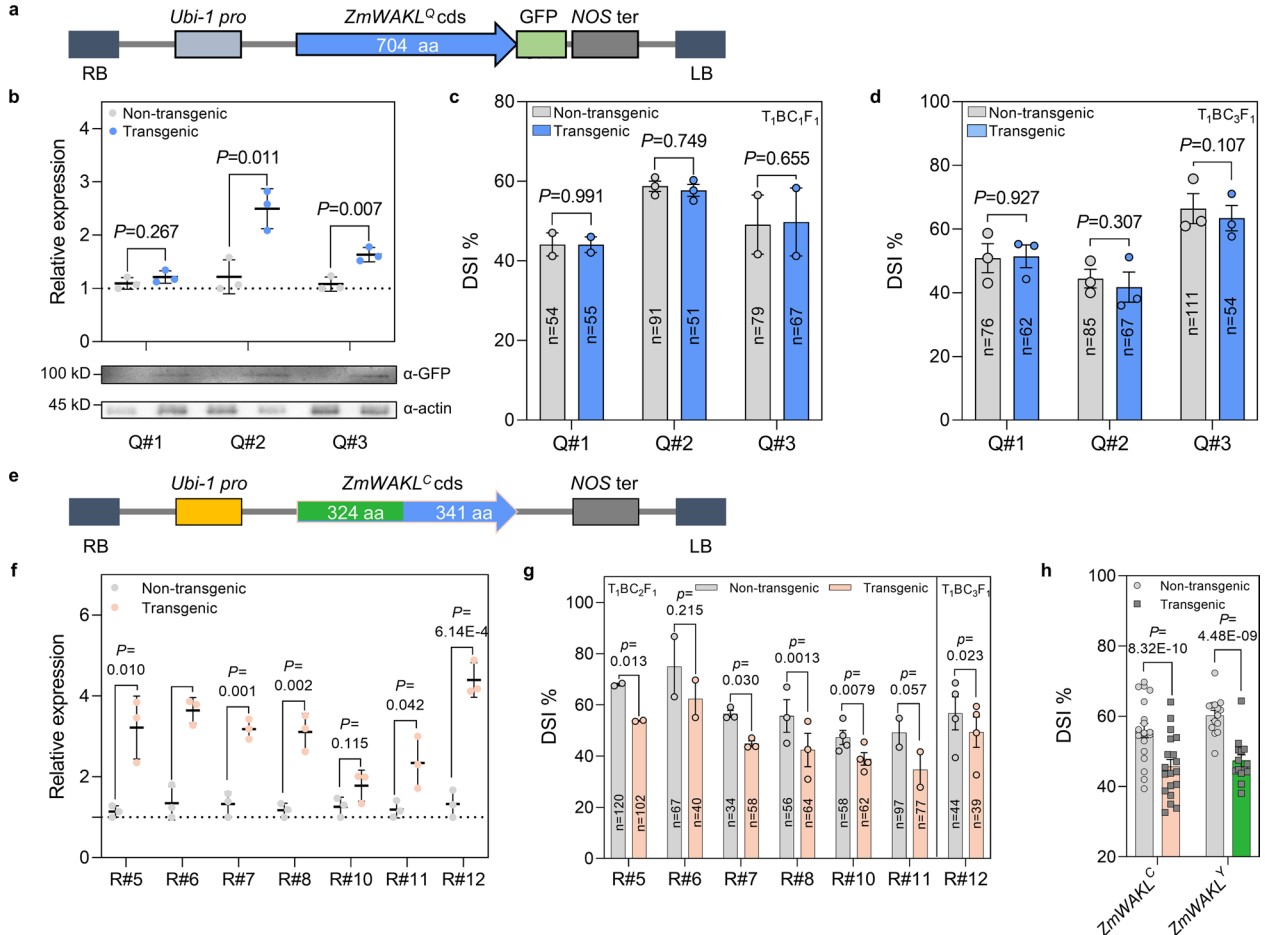

**Extended Data Fig. 4 | Transgenic functional validation of the susceptible**
***ZmWAKL^Q* allele and the chimeric *ZmWAKL^C* gene in GLS resistance.**
(**a**) Schematic diagram of the *Ubipro:ZmWAKL^Q-GFP* construct. (**b**) Relative
expression of *ZmWAKL* in *ZmWAKL^Q* overexpressing plants (n = 3). (**c,d**) GLS
resistance of the *ZmWAKL^Q* overexpression transgenic plants in backcross
populations (n = 397, T₁BC₁F₁; n = 455, T₁BC₃F₁). (**e**) Schematic diagram of the
*Ubipro:ZmWAKL^C* construct. The chimeric *ZmWAKL^C* gene was constructed
by combining the ZmWAKL^Y ECD/TM domains (1-324 aa) in-frame with the

ZmWAKL^Q intracellular domain (364-704 aa). (**f**) Relative expression of *ZmWAKL*
in *ZmWAKL^C* overexpressing plants (n = 3). (**g**) GLS resistance of the *ZmWAKL^C*
overexpression transgenic plants in backcross populations (n = 918). (**h**) DSI
values resulting from the overexpression of *ZmWAKL^Y* (n = 28) or *ZmWAKL^C*
(n = 38). Data are shown as means ± s.e. in (**c**), (**d**), (**g**) and (**h**); means ± s.d. in (**b**)
and (**f**). Statistical significance was determined by a two-sided Student's *t*-test in
(**b**) and (**f**) or a paired *t*-test in (**c**), (**d**), (**g**) and (**h**).

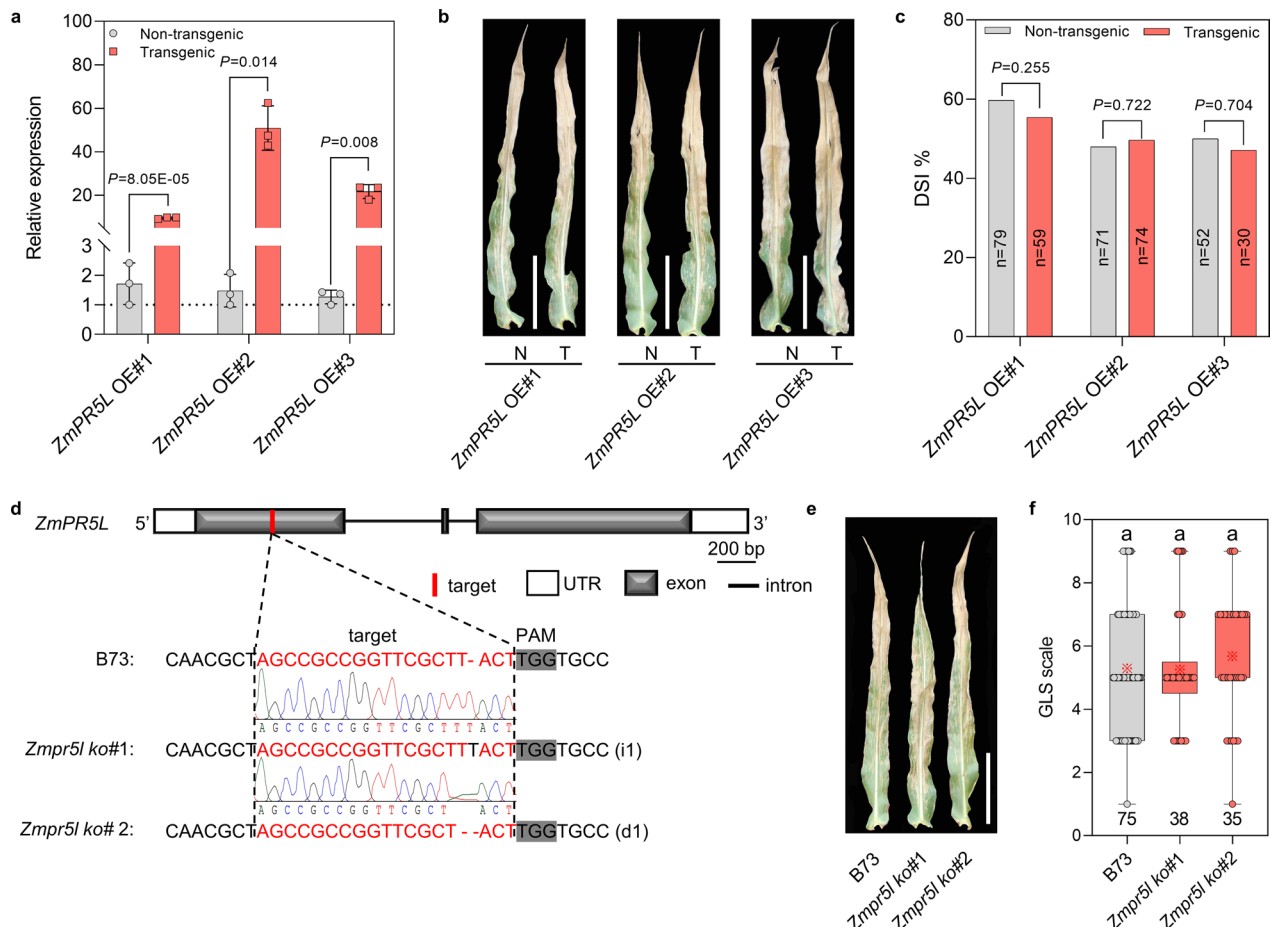

**Extended Data Fig. 5 | Transgenic functional validation of *ZmPR5L* in GLS resistance.** (**a**) Relative expression of *ZmPR5L* in plants overexpressing *ZmPR5L* (n = 3). (**b**,**c**) GLS symptoms (**b**) and DSI (**c**) in segregating T$_2$ transgenic plants (n = 365). N, non-transgenic plant; T, transgenic plant. Scale bars, 15 cm. (**d**) Schematic diagram of *ZmPR5L* and CRISPR/Cas9 editing target (red box). (**e**,**f**) GLS symptoms (**e**) and GLS scale (n = 148) (**f**) of the wild-type B73 and homozygous *Zmpr5l* knockout plants. Scale bar, 15 cm. Data are shown as means ± s.d. in (**a**); means in (**c**). Statistical significance was determined by a two-sided Student's *t*-test in (**a**) and (**c**). Data in (**f**) are displayed as box and whisker plots, and different lowercase letters indicate a significant difference (*P* < 0.05) based on one-way ANOVA with Fisher's least significant difference (LSD) test. The asterisk denotes the mean, and the box limits indicate the interquartile range.

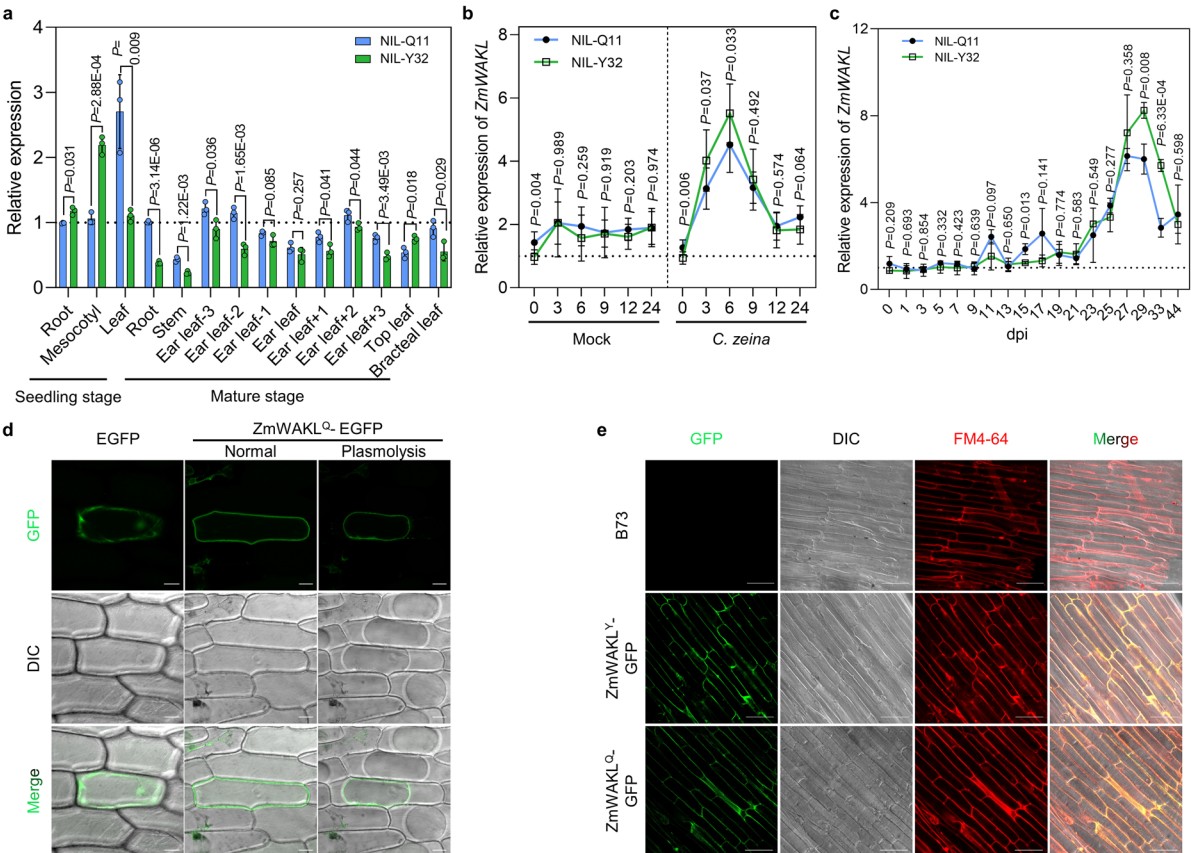

**Extended Data Fig. 6 | Molecular characterization of ZmWAKL.** (**a**) Relative *ZmWAKL* expression in tissues collected from NIL-Q11 and NIL-Y32 plants (n = 3). (**b,c**) Relative expression levels of *ZmWAKL* in NIL-Q11 and NIL-Y32 after inoculation with *C. zeina* in the greenhouse (**b**) or field (**c**). (**d,e**) Subcellular localization of ZmWAKL in onion epidermal cells (**d**) and *ZmWAKL-GFP* overexpression transgenic plants (**e**). Plasmolysis was achieved by incubating cells in 0.3 g/mL sucrose for 5 min. Scale bars, 50 μm. DIC, differential interference contrast. The membrane was stained with the membrane-impermeable dye FM4-64. This experiment was repeated two times with the same results. In (**a**–**c**), values are means ± s.d., statistical significance was determined by a two-sided Student's *t*-test.

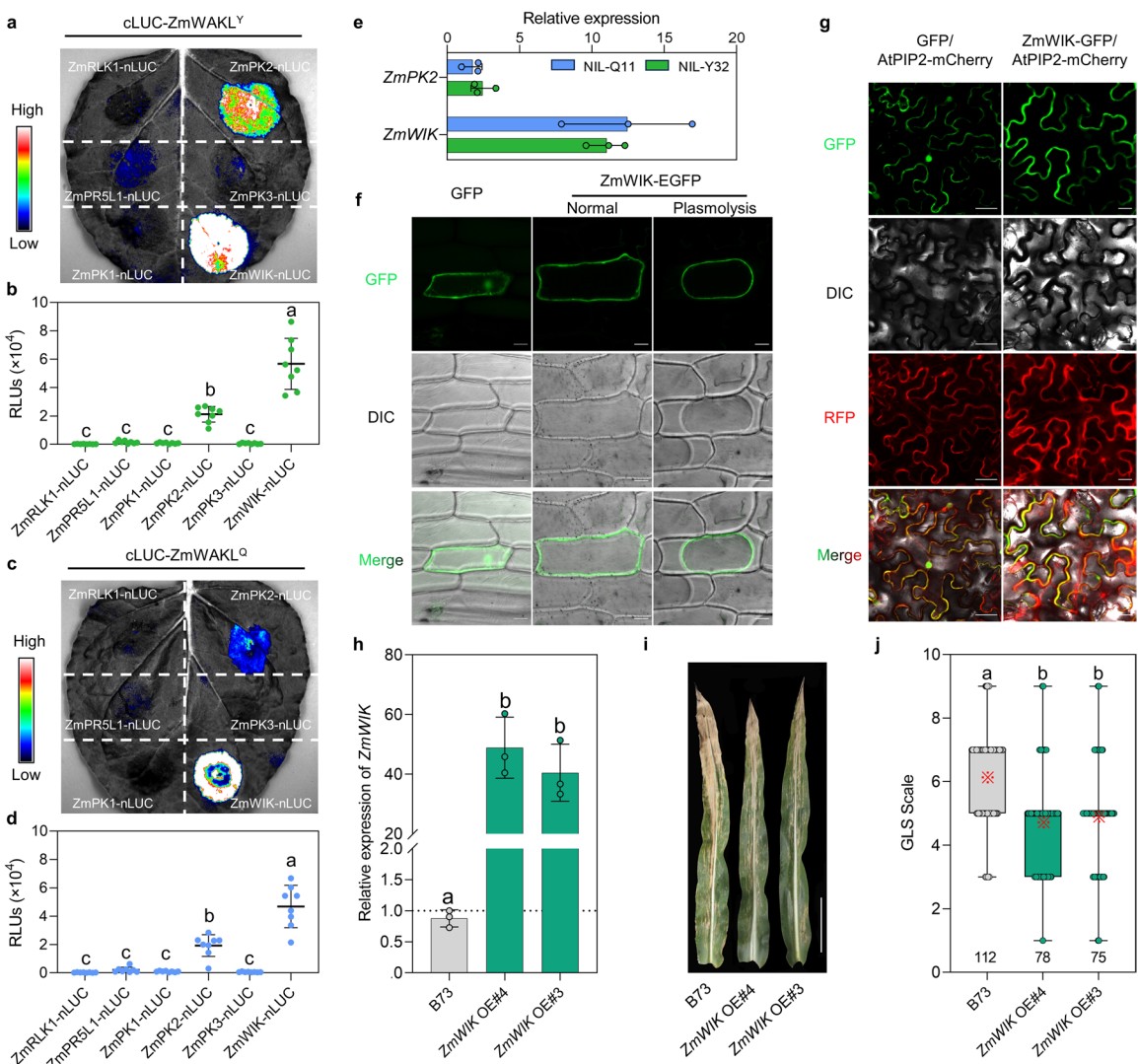

**Extended Data Fig. 7 | ZmWIK interacts with ZmWAKL$^{ICD}$ and regulates GLS resistance. (a,c)** Interactions of plasma membrane-tethered kinases with two ZmWAKL isoforms in SLC assays. **(b,d)** Analysis of relative luminescence units (RLUs) representing the interaction strengths between two ZmWAKL isoforms and plasma membrane-tethered kinases in pairwise combinations (n = 8). **(e)** Relative expression of *ZmPK2* and *ZmWIK* in leaves of NILs (n = 3). **(f,g)** Plasma-membrane localization of ZmWIK in onion epidermal cells **(f)** and *N. benthamiana* leaves **(g)**. Plasmolysis was achieved by incubating onion epidermal cells in 0.3 g/mL sucrose for 5 min. In *N. benthamiana* leaves, AtPIP2-mCherry (Plasma membrane intrinsic protein 2) was used as the membrane localization marker. DIC, differential interference contrast. Scale bars, 50 μm (onion epidermal cells); 100 μm (*N. benthamiana* leaves). This experiment was repeated two times with the same results. **(h)** Relative expression of *ZmWIK* in B73 and two homozygous overexpression transgenic lines (n = 3). **(i,j)** GLS symptoms **(i)** and scales (n = 265) **(j)** of B73 and transgenic lines. Scale bar, 15 cm. Data are shown as means ± s.d. in **(b)**, **(d)**, **(e)** and **(h)**, statistical significance was determined by a two-sided Student's *t*-test. Data in **(j)** are displayed as box and whisker plots with individual data points, the asterisk denotes the mean and the box limits indicate the interquartile range. In **(b)**, **(d)** and **(h)**, different lowercase letters indicate a significant difference (*P* < 0.05) based on one-way ANOVA with Tukey's test. In **(j)**, different lowercase letters indicate a significant difference (*P* < 0.05) based on one-way ANOVA with LSD test. The SLC assays were repeated at least four times.

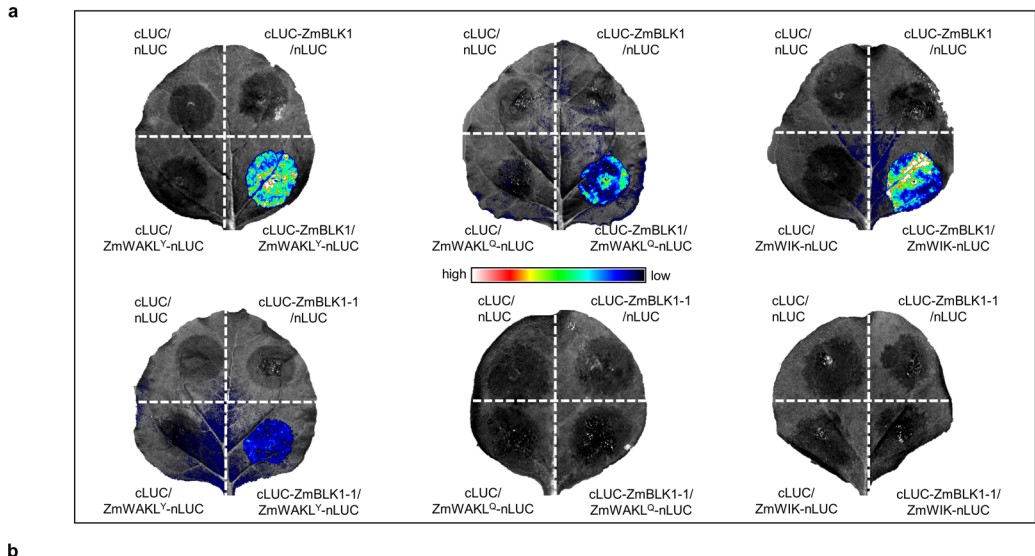

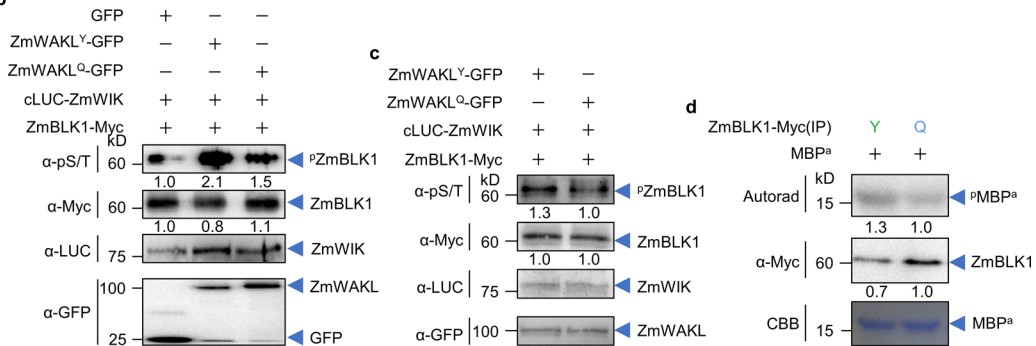

**Extended Data Fig. 8 | ZmBLK1 interacts with the ZmWAKL/ZmWIK complex for phosphorylation signaling. (a)** Determination of the interaction between ZmBLK1/ZmBLK1-1 and ZmWAKL or ZmWIK. **(b)** Phosphorylation of ZmBLK1 by ZmWAKL$^Y$/ZmWIK or ZmWAKL$^Q$/ZmWIK in *N. benthamiana* leaves. ZmBLK1 was immunoprecipitated with α-Myc magnetic beads (IP, α-Myc). The immunoprecipitated and input samples were immunoblotted with the indicated antibodies. α-pS/T, anti-phospho-(Ser/Thr) antibody. **(c)** Phosphorylation of ZmBLK1 by the mixed ZmWAKL$^Y$/ZmWIK and ZmWAKL$^Q$/ZmWIK *in vivo*. ZmBLK1 was immunoprecipitated with α-Myc magnetic beads (IP, α-Myc). The immunoprecipitated and input samples were immunoblotted with the

indicated antibodies. α-pS/T, anti-Phospho-(Ser/Thr) antibody. **(d)** Determination of ZmBLK1 kinase activity. The immunoprecipitated ZmBLK1 in **(c)** can phosphorylate the substrate MBP$^a$. Green 'Y' and blue 'Q' denote ZmBLK1 immunoprecipitated from the protoplasts of *Ubipro:ZmWAKL$^Y$-GFP* and *Ubipro:ZmWAKL$^Q$-GFP*, respectively. Phosphorylation of MBP$^a$ was detected by autoradiography (upper). Equal protein loading is shown by CBB staining (lower). The protein amounts were quantified by densitometry of band intensities, with the value to the right set to 1. The SLC assays were repeated at least two times and the *in vivo* kinase assays were repeated two times.

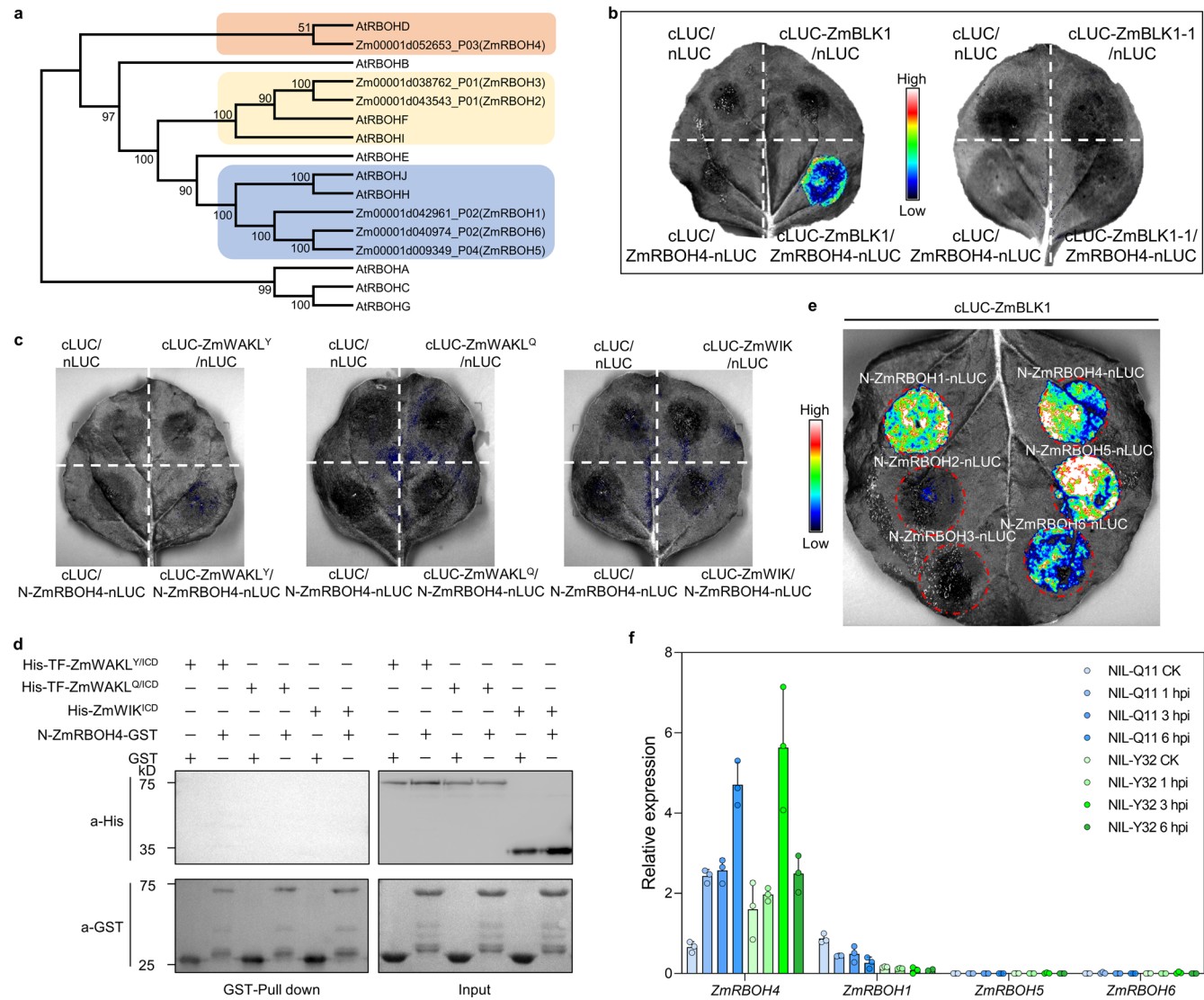

**Extended Data Fig. 9 | Investigation of the interactions between N-ZmRBOH4 and ZmWAKL (or ZmWIK). (a)** Phylogenetic analysis demonstrated that ZmRBOH4 is closely related to AtRBOHD. **(b)** Determination of the interaction between ZmBLK1/ZmBLK1-1 and ZmRBOH4. **(c,d)** N-ZmRBOH4 (1-377aa) does not interact with ZmWAKL[Y], ZmWAKL[Q] or ZmWIK in a SLC assay **(c)** or GST pull-down assay *in vitro* **(d)**. **(e)** Interactions of cLUC-ZmBLK1 with ZmRBOHs-nLUC in an SLC assay. **(f)** Relative expression levels of *ZmRBOHs* in NIL-Q11 and NIL-Y32 after inoculation with *C. zeina* (n = 3). hpi, hours post inoculation. The SLC and pull-down assays were repeated at least three times.

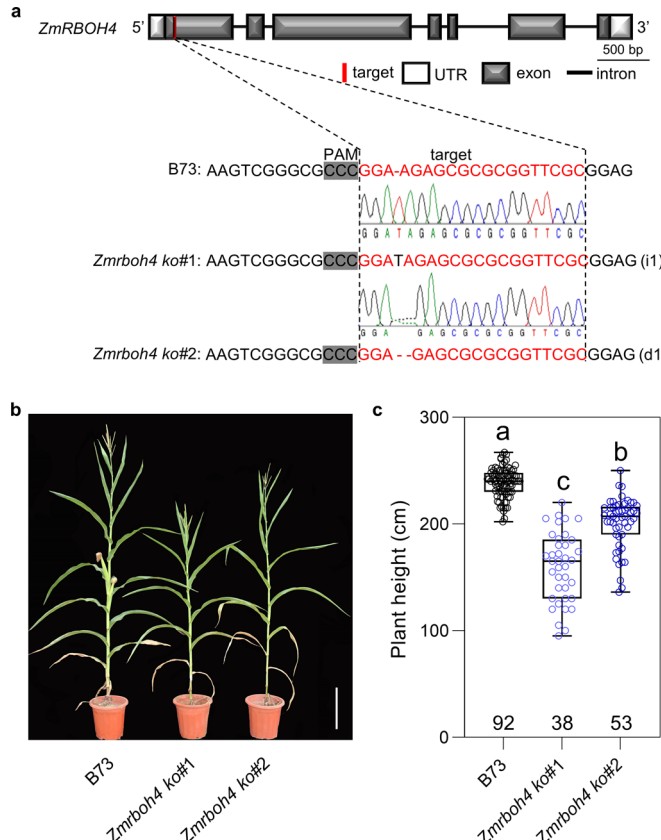

**Extended Data Fig. 10 | Knockout of *ZmRBOH4* by CRISPR/Cas9 results in changes to plant morphology.** (**a**) Schematic diagram of *ZmRBOH4* and the sgRNA target site in the first exon for CRISPR/Cas9-mediated editing. The sequences of two homozygous knockout lines with insertion (*ko* #1) and deletion (*ko* #2) are shown. The wild-type B73 sequence is shown on top. The target sites and PAM are highlighted in red and gray, respectively, with deletions indicated by dashes. (**b**,**c**) Plant morphology (**b**) and height (**c**) of B73 and two homozygous knockout lines (n = 183). The knockout lines exhibit significantly reduced height. Scale bar, 30 cm. Data in (**c**) are displayed as box and whisker plots with individual data points, the asterisk denotes the mean, and the box limits indicate the interquartile range. The center line indicates the median, and different lowercase letters indicate a significant difference (*P* < 0.05) based on one-way ANOVA with LSD test.

# Reporting Summary

## Statistics

For all statistical analyses, confirm that the following items are present in the figure legend, table legend, main text, or Methods section.

| n/a | Confirmed | |
|---|---|---|
| ☐ | ☒ | The exact sample size (*n*) for each experimental group/condition, given as a discrete number and unit of measurement |
| ☐ | ☒ | A statement on whether measurements were taken from distinct samples or whether the same sample was measured repeatedly |
| ☐ | ☒ | The statistical test(s) used AND whether they are one- or two-sided *Only common tests should be described solely by name; describe more complex techniques in the Methods section.* |
| ☒ | ☐ | A description of all covariates tested |
| ☒ | ☐ | A description of any assumptions or corrections, such as tests of normality and adjustment for multiple comparisons |
| ☐ | ☒ | A full description of the statistical parameters including central tendency (e.g. means) or other basic estimates (e.g. regression coefficient) AND variation (e.g. standard deviation) or associated estimates of uncertainty (e.g. confidence intervals) |
| ☐ | ☒ | For null hypothesis testing, the test statistic (e.g. *F*, *t*, *r*) with confidence intervals, effect sizes, degrees of freedom and *P* value noted *Give P values as exact values whenever suitable.* |
| ☒ | ☐ | For Bayesian analysis, information on the choice of priors and Markov chain Monte Carlo settings |
| ☒ | ☐ | For hierarchical and complex designs, identification of the appropriate level for tests and full reporting of outcomes |
| ☒ | ☐ | Estimates of effect sizes (e.g. Cohen's *d*, Pearson's *r*), indicating how they were calculated |

*Our web collection on statistics for biologists contains articles on many of the points above.*

## Software and code

Policy information about availability of computer code

| Data collection | The ImageJ 1.53c (https://imagej.net/ij/ij/download.html) was used to evaluate the gray value of western blotting and luminescence intensity. |
|---|---|
| Data analysis | For gene prediction, http://linux1.softberry.com/berry.phtml. For gene annotation, web based Blast2GO (https://www.blast2go.com). For predicting conserved domains, https://www.ncbi.nlm.nih.gov/Structure/cdd/wrpsb.cgi. For phylogenetic analysis, MEGA 7.0 (https://www.megasoftware.net/). For nucleotide diversity, ClustalX2 (http://www.clustal.org/) was used to produce a nucleotide alignment matrix, and DnaSP6 (http://www.ub.edu/dnasp/DnaSP_OS.html) was used for nucleotide diversity (π) analysis, Tajima's D test and haplotype analysis. For statistical analysis, performed by IBM SPSS Statistics SV26 (https://www.ibm.com/products/spss-statistics). For data visualization, GraphPad Prism 8 (https://www.graphpad.com/). |

For manuscripts utilizing custom algorithms or software that are central to the research but not yet described in published literature, software must be made available to editors and reviewers. We strongly encourage code deposition in a community repository (e.g. GitHub). See the Nature Portfolio guidelines for submitting code & software for further information.

## Data

Policy information about availability of data

All manuscripts must include a data availability statement. This statement should provide the following information, where applicable:
- Accession codes, unique identifiers, or web links for publicly available datasets
- A description of any restrictions on data availability
- For clinical datasets or third party data, please ensure that the statement adheres to our policy

The authors declare that the data supporting the findings of this study are available within the paper and its supplementary information files. The reported WAKs/WAKLs and RLCKs' protein sequences are downloaded from the National Center for Biotechnology Information database (NCBI, http://www.ncbi.nlm.nih.gov/). The protein sequences of AtRBOHs are downloaded from The Arabidopsis Information Resource database (TAIR, https://www.arabidopsis.org/), and the protein sequences of ZmRBOHs are obtained from the Gramene database (https://www.gramene.org/). The expression data of ZmRBOHs and receptor-like kinases is obtained from the Plant Public RNA-seq Database (http://ipf.sustech.edu.cn/pub/plantrna/). The B73 genomic sequences in the mapped qRgls1 region are collected from the MaizeGDB (https://www.maizegdb.org/). The sequences of two BAC clones (17-37-1-53 and 57-9-1-93) are available at GenBank accessions OQ435908 and OQ435909, respectively. The genomic sequences of ZmWAKL used for haplotype analysis are available at GenBank under accessions OQ425304-OQ425401. The coding sequences of ZmWAKLY and ZmWAKLQ are available at GenBank accessions OQ421108 and OQ421109, respectively. The genomic and coding sequences of ZmPR5Y and ZmPR5Q are available at GenBank accessions OQ421106-OQ421107 and OQ421110-OQ421111, respectively. The coding sequence of ZmWIK is available at GenBank accession OQ421112. The coding sequences of ZmBLK1 and ZmBLK1-1 are available at GenBank accessions OQ421113 and OQ421114, respectively. The N-terminal of ZmRBOH4 is available at GenBank accession OQ421115. Source data are provided with this paper.

## Human research participants

Policy information about studies involving human research participants and Sex and Gender in Research.

| Reporting on sex and gender | NA |
|---|---|
| Population characteristics | NA |
| Recruitment | NA |
| Ethics oversight | NA |

Note that full information on the approval of the study protocol must also be provided in the manuscript.

# Field-specific reporting

Please select the one below that is the best fit for your research. If you are not sure, read the appropriate sections before making your selection.

☒ Life sciences  ☐ Behavioural & social sciences  ☐ Ecological, evolutionary & environmental sciences

For a reference copy of the document with all sections, see nature.com/documents/nr-reporting-summary-flat.pdf

# Life sciences study design

All studies must disclose on these points even when the disclosure is negative.

| Sample size | Sample size are indicated in individual figures and figure legends. The sample size for homozygous lines with/without the Y32 fragment is 1151 plants, while the fine-mapping populations consist of 3290 plants. For the NIL's flowering-related traits, there are 5 repeats with 195 plants in Beijing, and 3 repeats with 96 plants in Hainan; for the transgenic materials' flowering-related traits, the complementary materials have 3 repeats with 235 plants, and the overexpression materials have 3 repeats with 353 plants. In transgenic verification experiments, the sample sizes are as follows: 790 plants for complementation assays, 1765 plants for ZmWAKLY overexpression, 852 plants for ZmWAKLQ overexpression, 918 plants for ZmWAKLC overexpression, and for ZmPR5L overexpression and CRISPR/Cas9 knockout, 365 and 148 plants respectively. ZmWIK overexpression and mutation lines have 265 and 132 plants, respectively. Lastly, the ZmRBOH4 knockout involves 183 plants. For RT-qPCR, each leaf tissue had three samples, and each sample was harvested from three plants. RNA expression for each sample was tested with three technical replicates. For oxidative burst assay, each sample had two strips with at least 6 replicates. For the protein content and interaction strength analysis, 8 group samples were used for testing. Statistical between two groups were analyzed by two-sided student's t-test or paired t-test. Statistical significance between more than two groups were analyzed based on one-way ANOVA with Tukey's test or Fisher's least significant difference (LSD) test, different lowercase letters indicate a significant difference (P < 0.05). |
|---|---|
| Data exclusions | No data were excluded from the analyses. |
| Replication | The phenotype of recombinants and NILs were identified over several years with the similar results. In the transgenic verification assays, we have identified the phenotype of transgenic materials for more than 2 years and tested the gene's function in different genetic backgrounds, and similar results were obtained across different years and generations. If we have sufficient materials, we set up at least 3 replicates every year. For the flowering-related traits experiments of NILs, we set up two locations in Beijing and Hainan, with 5 repeats in Beijing and 3 repeats in Hainan; for the transgenic flowering-related assays, 3 repeats were set up. In the split-luciferase complementation assays, more |

than three leaves (three replicates) were used for each assay. In other protein interaction assays (pull down, co-immunoprecipitation and SLC), at least three replicates were set up for each experiment. For the in vitro and in vivo phosphorylation assay, we repeated at least two times. All the replications have the similar results, and all the attempts were successful.

| | |
|---|---|
| Randomization | Briefly, the samples were allocated into experimental groups randomly. In the fine-mapping of qRgls1, as all the recombinant-derived progeny was grown in the same plot, therefore, all individuals, with or without qRgls1, were randomly distributed in the same experimental plot. All the recombinants and replications were randomly planted in the field. For the transgenic verification assays, we adopted the same strategy. The segregating progeny of a transgenic event were randomly distributed in the same plot and the same transgenic event was randomly planted in the field. The homozygous transgenic plants and transgenic receptor lines were planted together and the replications were randomly distributed in the fields. |
| Blinding | The investigators were blinded to group allocation during data collection and analysis. |

# Reporting for specific materials, systems and methods

We require information from authors about some types of materials, experimental systems and methods used in many studies. Here, indicate whether each material, system or method listed is relevant to your study. If you are not sure if a list item applies to your research, read the appropriate section before selecting a response.

## Materials & experimental systems

| n/a | Involved in the study |
|---|---|
| ☐ | ☒ Antibodies |
| ☒ | ☐ Eukaryotic cell lines |
| ☒ | ☐ Palaeontology and archaeology |
| ☒ | ☐ Animals and other organisms |
| ☒ | ☐ Clinical data |
| ☒ | ☐ Dual use research of concern |

## Methods

| n/a | Involved in the study |
|---|---|
| ☒ | ☐ ChIP-seq |
| ☒ | ☐ Flow cytometry |
| ☒ | ☐ MRI-based neuroimaging |

## Antibodies

| | |
|---|---|
| Antibodies used | Antibody, supplier name, catalog number, clone name, lot number.<br>1. Mouse anti GFP-Tag mAb, ABclonal, #AE012, AMC0507, 9200012003;<br>2. Anti-Plant-actin Mouse Monoclonal Antibody, EASYBIO, #BE0028, Q30, 80870207;<br>3. Anti-Firefly Luciferase antibody (Goat), abcam, #ab181640, NA, GR3267949-18;<br>4. Mouse anti Myc-Tag mAb, ABclonal, #AE010, AMC0048, 3500014031;<br>5. Mouse anti MBP-Tag mAb, ABclonal, #AE016, AMC0505, 9200016002;<br>6. HRP-conjugated GST-tag Mouse mAb, YEASEN, #30903ES10, 3C10, H7227380;<br>7. HRP-conjugated His-tag Mouse mAb, YEASEN, #30404ES60, 9C11, H6227090;<br>8. Anti-Phospho-(Ser/Thr) antibody (Rabbit), abcam, #ab117253, NA, GR3376370-6;<br>9. Goat Anti-Rabbit IgG (H&L)-HRP Conjugated, EASYBIO, #BE0101, NA, 80780926;<br>10. Goat Anti-Mouse IgG (H&L)-HRP Conjugated, EASYBIO, #BE0102, NA, 80781014;<br>11. Rabbit Anti-Goat IgG (H&L)-HRP Conjugated, EASYBIO, #BE0103, NA, 80921201. |
| Validation | 1. Mouse anti GFP-Tag mAb<br>WB: Nicotiana benthamiana (Manufacturer's websit: https://abclonal.com.cn/catalog/AE012), Zea mays (this study);<br>Co-IP: Nicotiana benthamiana (this study);<br>Reference: Yang Z, Huang Y, Yang J, Yao S, Zhao K, Wang D, Qin Q, Bian Z, Li Y, Lan Y, Zhou T, Wang H, Liu C, Wang W, Qi Y, Xu Z, Li Y. Jasmonate Signaling Enhances RNA Silencing and Antiviral Defense in Rice. Cell Host Microbe. 2020 Jul 8;28(1):89-103.e8. doi: 10.1016/j.chom.2020.05.001. PMID: 32504578;<br>Manufacturer's specification: https://abclonal.com.cn/Datasheet/Antibodies/AE012.pdf?v=1669714209.<br><br>2. Anti-Plant-actin Mouse Monoclonal Antibody<br>WB: Various Plant (Manufacturer's websit: http://www.bioeasytech.com/product/2363.html?goods_id=4251);<br>Specificity: Antibody can detects endogenous plant actin protein.<br><br>3. Anti-Firefly Luciferase antibody (Goat)<br>Suitable for WB: Nicotiana benthamiana (Manufacturer), Zea mays (this study)<br>Reference: Liu C, Cui D, Zhao J, Liu N, Wang B, Liu J, Xu E, Hu Z, Ren D, Tang D, Hu Y. Two Arabidopsis Receptor-like Cytoplasmic Kinases SZE1 and SZE2 Associate with the ZAR1-ZED1 Complex and Are Required for Effector-Triggered Immunity. Mol Plant. 2019 Jul 1;12(7):967-983. doi: 10.1016/j.molp.2019.03.012. PMID: 30947022;<br>The Key features and eatails are on Manufacturer's websit: https://www.abcam.cn/firefly-luciferase-antibody-ab181640.html?productWallTab=Abreviews.<br><br>4. Mouse anti Myc-Tag mAb<br>WB: Zea mays, Nicotiana benthamiana (Manufacturer's websit: https://abclonal.com.cn/catalog/AE010);<br>Co-IP: Nicotiana benthamiana (this study);<br>Reference: Yang Z, Huang Y, Yang J, Yao S, Zhao K, Wang D, Qin Q, Bian Z, Li Y, Lan Y, Zhou T, Wang H, Liu C, Wang W, Qi Y, Xu Z, Li Y. Jasmonate Signaling Enhances RNA Silencing and Antiviral Defense in Rice. Cell Host Microbe. 2020 Jul 8;28(1):89-103.e8. doi: |

10.1016/j.chom.2020.05.001. PMID: 32504578;
Manufacturer's specification: https://abclonal.com.cn/Datasheet/Antibodies/AE010.pdf?v=1675416399.

5. Mouse anti MBP-Tag mAb
Manufacturer's websit: https://abclonal.com.cn/catalog/AE016;
Refeence: Xiong J, Yang F, Yao X, Zhao Y, Wen Y, Lin H, Guo H, Yin Y, Zhang D. The deubiquitinating enzymes UBP12 and UBP13 positively regulate recovery after carbon starvation by modulating BES1 stability in Arabidopsis thaliana. Plant Cell. 2022 Oct 27;34(11):4516-4530. doi: 10.1093/plcell/koac245. PMID: 35944221;
Manufacturer's specification: https://abclonal.com.cn/Datasheet/Antibodies/AE016.pdf?v=1666064798.

6. HRP-conjugated GST-tag Mouse mAb, YEASEN, #30903ES10, NA, NA;
Application: WB (https://www.yeasen.com/products/detail/170);
Manufacturer's specification: https://upload.yeasen.com/website/file/20180830123807687.pdf.

7. HRP-conjugated His-tag Mouse mAb
Application: WB (https://www.yeasen.com/products/detail/86);
Reference: Tan FQ, Wang W, Li J, Lu Y, Zhu B, Hu F, Li Q, Zhao Y, Zhou DX. A coiled-coil protein associates Polycomb Repressive Complex 2 with KNOX/BELL transcription factors to maintain silencing of cell differentiation-promoting genes in the shoot apex. Plant Cell. 2022 Jul 30;34(8):2969-2988. doi: 10.1093/plcell/koac133. PMID: 35512211;
Manufacturer's specification: https://upload.yeasen.com/website/file/20180827222215846.pdf.

8. Anti-Phospho-(Ser/Thr) antibody
WB application in Reference: Peng X, Wang M, Li Y, Yan W, Chang Z, Chen Z, Xu C, Yang C, Deng XW, Wu J, Tang X. Lectin receptor kinase OsLecRK-S.7 is required for pollen development and male fertility. J Integr Plant Biol. 2020 Aug;62(8):1227-1245. doi: 10.1111/jipb.12897. PMID: 31833176 [Protein phosphorylation was detected by immunoblotting with anti-phospho Ser/Thr antibodies (Abcam, Cat#ab117253)]; Nicotiana benthamiana, Zea mays (this study);
The Key features and eatails are on Manufacturer's websit: https://www.abcam.cn/phospho-serthr-antibody-ab117253.html.

9. Goat Anti-Rabbit IgG (H&L)-HRP Conjugated
Application: WB (http://www.bioeasytech.com/product/2901.html?goods_id=5786).

10. Goat Anti-Mouse IgG (H&L)-HRP Conjugated
Application: WB (http://www.bioeasytech.com/product/2907.html?goods_id=5794);
Reference: Liu Q, Deng S, Liu B, Tao Y, Ai H, Liu J, Zhang Y, Zhao Y, Xu M. A helitron-induced RabGDIα variant causes quantitative recessive resistance to maize rough dwarf disease. Nat Commun. 2020 Jan 24;11(1):495. doi: 10.1038/s41467-020-14372-3. PMID: 31980630.

11. Rabbit Anti-Goat IgG (H&L)-HRP Conjugated
Application: WB (http://www.bioeasytech.com/product/2914.html?goods_id=5803).

