## [Peer Review File · Nature Genetics]

Peer Review Information

Manuscript Title: The ZmWAKL-ZmWIK-ZmBLK1-ZmRBOH4 module provides quantitative resistance to gray leaf spot in maize

Corresponding author name(s): Professor Mingliang Xu

Reviewer Comments & Decisions:

Decision Letter, initial version:
--

24th Mar 2023

Dear Professor Xu,

Your Article, "ZmWAKL-mediated immunity underlies quantitative disease resistance to gray leaf spot in maize" has now been seen by 3 referees. You will see from their comments below that while they find your work of interest, some important points are raised. We are interested in the possibility of publishing your study in Nature Genetics, but would like to consider your response to these concerns in the form of a revised manuscript before we make a final decision on publication.

To guide the scope of the revisions, the editors discuss the referee reports in detail within the team with a view to identifying key priorities that should be addressed in revision. In this case, we think all three referees have provided constructive reviews aimed at strengthening the experimental evidence to support some of the conclusions, and improving the presentation and discussion. We particularly ask that you address their comments as thoroughly as possible with appropriate revisions. We hope that you will find the prioritized set of referee points to be useful when revising your study. Please do not hesitate to get in touch if you would like to discuss these issues further.

We therefore invite you to revise your manuscript taking into account all reviewer and editor comments. Please highlight all changes in the manuscript text file. At this stage we will need you to upload a copy of the manuscript in MS Word .docx or similar editable format.

*1) Include a "Response to referees" document detailing, point-by-point, how you addressed each

referee comment. If no action was taken to address a point, you must provide a compelling argument. This response will be sent back to the referees along with the revised manuscript.

*2) If you have not done so already please begin to revise your manuscript so that it conforms to our Article format instructions, available [here](http://www.nature.com/ng/authors/article_types/index.html). Refer also to any guidelines provided in this letter.

[redacted]

We hope to receive your revised manuscript within 3 to 6 months. If you cannot send it within this time, please let us know.

Sincerely,
Wei

Wei Li, PhD

Senior Editor
Nature Genetics
New York, NY 10004, USA
www.nature.com/ng

Reviewers' Comments:

Reviewer #1:

Remarks to the Author:

Zhong et al. describe an extensive study that set out to identify the gene(s) underlying the qRgls1 QTL in maize that confers quantitative disease resistance to gray leaf spot (caused by *Cercospora zeina*). Using positional mapping, they delineate the region to a 122 kb region in the reference genome. Sequencing of BAC clones in the resistant donor found a region of 60kb with two candidate genes in the region. The authors functionally identify ZmWAKL as the gene underlying qRgls1 using transgenic complementation using native and overexpression. Further work delineated the role of the divergent extracellular domain as determining the specificity of recognition. The manuscript pivots from genetics to biochemistry/cell biology wherein the authors show that:

1. ZmWAKL is membrane associated
2. ZmWAKL has substantial variation in its ectodomain
3. Divergence in resistant/susceptible alleles of ZmWAKL for self-association and kinase activity
4. Identify ZmWIK as the co-receptor of ZmWAKL (using IP-MS)
5. Use a large number of assays to show the phosphorylation events between these two proteins in vitro and in vivo
6. Identification of an receptor-like cytoplasmic kinase (RLCK), here ZmBLK1, as a phosphorelay to maize ortholog of RBOHD.

While similar phenomena have been well described in *Arabidopsis thaliana* in the regulation of the pattern recognition receptors FLS2 and EFR, little work has occurred in grass species. Beyond the fundamental and agricultural importance of the discovery that a wall-associated transmembrane kinase underlies the qRgls1 QTL, this work demonstrates within a single manuscript the complex formation and phosphocode that relays a signal to RBOHD. While I agree with the authors interpretation that this signal leads to RBOHD, the paper does not demonstrate conclusively that RBOHD represents the only pathway for this immune signaling (for example, a knockout of RBOHD may have pleiotropic effects that leads to increased susceptibility). I mention this last comment as the only real criticism of the paper and would encourage the authors in the discussion to entertain additional roles of signal transduction and immune pathways that would contribute to resistance.

Methods/Approach comments

The methods do not describe the approach use for multiple sequence alignment, how many bootstraps were performed for phylogenetic inference, and whether domains or full-length sequence were used in the alignment. This needs to be resolved in the methods and comments in figure legends.

Figure comments

Figure 1. Include information about the backcrossing information for panels c-o in the figure description. Review color coding information, I don't think the colors in panel C are described. Figure 4c, Co-IP is only showing association. Interaction can often be construed as a direct interaction

as compared to an indirect interaction. Similar language has been used with assays that only show proximity but not interaction.

Throughout figure legends, the number of times that experiments (specifically co-IPs, luciferase assays, and phosphorylation assays) are not consistently mentioned.

Extended Figure 8, panel b does not really have much meaning without bootstrap support.

Minor comments

Throughout the manuscript, 'Extended Data' is misspelled.

'Extended Data Figures' is misspelled

Did the transgenes include introns or not? Through the use of a large arrow, the figures give the impression that they did not.

Reviewer #2:

Remarks to the Author:

GLS is a destructive disease on maize. And very few resistant genes against GLS were cloned. This manuscript described the cloning of ZmWAKL, a resistant gene against GLS by map-based cloning. Based on the disease evaluation of transgenic plants, it was confirmed that ZmWAKL gene can enhance maize resistance against GLS. Further, the authors uncovered the ZmWAKL-ZmWIK-ZmBLK1-ZmRBOH4 module which mediates resistance against GLS. This work is excellent.

Major comments:

1. No solid evidences were provided to confirm the genetic relationships among ZmWAKL, ZmWIK, ZmBLK1 and ZmRBOH4 in maize plants. Although the authors did a lot of in vitro phosphorylation assays and SLC assays, but those results are not enough to confirm their genetic relationship in maize plants. I strongly suggest the authors do some crosses by using transgenic maize plants and mutants to examine the relationship among the four resistant genes. For example, the authors can examine the relationship between ZmWAKL and ZmWIK by checking the disease phenotypes of ZmWAKL/zmwik, Ubi::ZmWIK/zmwakl, ZmWAKL, zmwik mutant and transgenic wild type plants. Also, the authors mentioned that ZmWAKL and ZmWIK phosphorylate each other (Line 321). This means that ZmWIK might work on the upstream of ZmWAKL protein. So, it is required to confirm the relationship between ZmWAKL and ZmWIK, between ZmWIK and ZmBLK1, between ZmBLK1 and ZmRBOH4.
2. Since phosphorylation of ZmWAKL, ZmWIK, ZmBLK1 and ZmRBOH4 are critical for the signal transduction, I strongly suggest the authors check whether the phosphorylation levels of those proteins are induced in transgenic maize plants by *C. zeina* infection. Also, it is better to check whether the enhanced phosphorylation levels of ZmBLK1 and ZmRBOH4 induced by *C. zeina* infection are dependent on ZmWAKLY by using transgenic plants.
3. Doses introgression of ZmWAKLY allele into susceptible maize lines affect their yield in GLS region and non-GLS region?
4. Scientific writing should be comprehensively improved in general.
5. Since GLS can be caused by *C. zeina* and *C. zea-maydis*, did the authors test whether ZmWAKL contributes resistance against *C. zea-maydis*?
6. Line 35, please delete "unidirectional". No solid evidences were provided to confirm it. The author mentioned that ZmWAKL and ZmWIK phosphorylate each other. So, they regulate each other. In Arabidopsis FLS2-BAK1-BIK1 complex, flg22-activated FLS2 protein phosphorylates BAK1, then BAK1 phosphorylates BIK1. And, BAK1 and FLS2 are substrates of BIK1 (PNAS, 2010, 107:496-501). It is

possible the ZmWAKL-ZmWIK-ZmBLK1 complex, similar as FLS2-BAK-BIK1 complex, is complicated and those subunits phosphorylate and regulate each other.

Reviewer #3:

Remarks to the Author:

The manuscript by Zhong et al describes the fine mapping and cloning of a ZmWAKL gene underlying the major quantitative resistance against gray leaf spot in maize. The WAK gene family has previously identified as quantitative resistant genes in several important crops and thus, these receptors are of high relevance for plant resistance to pathogen infection. The authors discovered the self-association of ZWAKL and identified its co-receptor WIK. By this, the authors discovered a WAKL/WIK-BLK-RBOH4 phosphorylation cascade and revealed the WAKL transduced ROS production mechanism against *C. Zeina* infection. The molecular experiments results are complete and convincing. Moreover, further genetic evidence is provided by a transgenic assay in maize. This presented work will significantly improve our understanding on quantitative resistant mechanisms and therefore is of high scientific value. The manuscript is well written and all over it was a pleasure to read this work.

There are some minor points I would suggest to modify or include:

1. Are there any variations of WIK, BLK and ZmRBOH4 between Y and Q? In addition to that, several transgenic mutants were made in the B73 genetic background, what's the difference between B73, Y and Q alleles of these genes
2. In line 214-215, authors concluded that the induction pattern of WAKL is consistent with the infection process of *C. Zeina*. Is this a part of a not-shown result, or a citation from (missing) references?
3. The luciferase signal is not a fluorescence signal. The authors should correct this terminology throughout the paper.
4. Line 320, the EMS or CRISPR knockout is not a deletion mutant.
5. In line 321-322, the authors need to explain which result support the conclusion of "more frequent phosphorylation is made by ZmWAKLY to ZmWIK".
6. BLK is an important immune signaling component and RBOH4 is crucial for ROS burst. Knock out the such genes could result in the reduction of basal immunity, thus causing the mutant being more susceptible to several disease but not dependent to WAK mediated resistance. The author should include this possibility in the discussion.
7. As the authors mention in the introduction, WAK can also recognize pathogen effectors. Since the elicitor of WAKL is unknown, the author should not sue "WAK/WAKL-related PTI mechanism" in the line 575.
8. In the methods section - what is the concentration of spores of *C. zeina* inoculum used for field assays?

Author Rebuttal to Initial comments
--

Reviewers' Comments:

Reviewer #1:

Remarks to the Author:

Zhong et al. describe an extensive study that set out to identify the gene(s) underlying the qRgls1 QTL in maize that confers quantitative disease resistance to gray leaf spot (caused by *Cercospora zeina*). Using positional mapping, they delineate the region to a 122 kb region in the reference genome. Sequencing of BAC clones in the resistant donor found a region of 60kb with two candidate genes in the region. The authors functionally identify ZmWAKL as the gene underlying qRgls1 using transgenic complementation using native and overexpression. Further work delineated the role of the divergent extracellular domain as determining the specificity of recognition. The manuscript pivots from genetics to biochemistry/cell biology wherein the authors show that:

1. ZmWAKL is membrane associated
2. ZmWAKL has substantial variation in its ectodomain
3. Divergence in resistant/susceptible alleles of ZmWAKL for self-association and kinase activity
4. Identify ZmWIK as the co-receptor of ZmWAKL (using IP-MS)
5. Use a large number of assays to show the phosphorylation events between these two proteins in vitro and in vivo
6. Identification of a receptor-like cytoplasmic kinase (RLCK), here ZmBLK1, as a phosphorelay to maize ortholog of RBOHD. While similar phenomena have been well described in *Arabidopsis thaliana* in the regulation of the pattern recognition receptors FLS2 and EFR, little work has occurred in grass species. Beyond the fundamental and agricultural importance of the discovery that a wall-associated transmembrane kinase underlies the qRgls1 QTL, this work demonstrates within a single manuscript the complex formation and phosphocode that relays a signal to RBOHD. While I agree with the authors interpretation that this signal leads to RBOHD, the paper does not demonstrate conclusively that RBOHD represents the only pathway for this immune signaling (for example, a knockout of RBOHD may have pleiotropic effects that leads to increased susceptibility). I mention this last comment as the only real criticism of the paper and would encourage the authors in the discussion to entertain additional roles of signal transduction and immune pathways that would contribute to resistance.

Response: We greatly appreciate the reviewer's very positive comments on our manuscript and would like to take this opportunity to express our gratitude for the constructive suggestions, which have helped us to enhance the clarity and comprehensiveness of our study. As known, the plant immune system is multilayered and complex, controlled by numerous genes. In our initial QTL mapping, we detected two major and two minor resistance QTL against gray leaf spot. In the current study, we report the map-based cloning of one major QTL *qRgls1* and its underlying mechanism against GLS. Our research has revealed the involvement of the ZmWAKL-ZmWIK-ZmBLK1-ZmRBOH4 immune module in initiating immune signaling triggered by *C. zeina* and eliciting a ROS burst to defend against GLS disease. Nonetheless, we acknowledge that at this stage, we cannot conclusively claim that RBOHD represents the only pathway for this immune signaling. While our knockout of RBOHD resulted in reduced ROS accumulation, which is closely associated with reduced GLS resistance, we must remain open to the possibility of other contributing factors. As the reviewer pointed out, potential pleiotropic effects could influence the overall susceptibility to GLS. In the revised version, we provide an explanation in the discussion section to address this aspect. Please refer to lines 587-590.

Methods/Approach comments

The methods do not describe the approach use for multiple sequence alignment, how many bootstraps were performed for phylogenetic inference, and whether domains or full-length sequence were used in the alignment.

This needs to be resolved in the methods and comments in figure legends.

Response: Thank you for your comment. In response to your suggestion, we have added bootstrap support and sequence information in the methods section. Please refer to lines 912-915 in the revised version.

Figure comments

Figure 1. Include information about the backcrossing information for panels c-o in the figure description. Review color coding information, I don't think the colors in panel C are described.

Response: As suggested, we add the backcrossing information in the legend of the revised Figure 1. Please refer to lines 1134 in the revised version. We apologize for our carelessness in confusing you. In panels c-e, they represent the same set of materials, with each column corresponding to a single material across panel c to panel e. For the sake of conciseness, we have only marked the materials' name at the bottom of panel e.

Figure 4c, Co-IP is only showing association. Interaction can often be construed as a direct interaction as compared to an indirect interaction. Similar language has been used with assays that only show proximity but not interaction. Throughout figure legends, the number of times that experiments (specifically co-IPs, luciferase assays, and phosphorylation assays) are not consistently mentioned. Extended Figure 8, panel b does not really have much meaning without bootstrap support.

Response: Thank you for the comments. Indeed, Co-IP assays can only indicate associations between proteins and not direct interactions. Typically, we employ multiple *in vivo* and *in vitro* assays to ascertain protein-protein interactions. We have made the necessary modifications in the text to accurately describe these protein associations/interactions. For specific details, please refer to lines 364-366 in the revised version. We appreciate the reviewer's reminder, and as suggested, we have included the number of replicates for different experiments in figure legends. In the methods, we have included the bootstrap support in the phylogenetic analysis. Please refer to lines 914-915 in the revised version.

Minor comments

Throughout the manuscript, 'Extended Data' is misspelled.

Response: Sorry for the typing error. We have corrected it throughout the manuscript.

'Extended Data Figures' is misspelled.

Response: Thank you, it has been corrected.

Did the transgenes include introns or not? Through the use of a large arrow, the figures give the impression that they did not.

Response: Thank you for the comments. In the functional complementation test, we subcloned the intact native *ZmWAKL* gene, including 5'-promoter, exons, introns, and 3'-end sequence, from the Y32 BAC clone to construct the vector (Figures 1a and 1b). The detailed gene structure is depicted in Extended data Fig. 2f. For the overexpression vectors, we cloned the coding sequence (cds) into the expression vector driven by *Ubiquitin* promoter. To distinguish between these two different vectors, we marked the native *ZmWAKL* allele (where boxes and lines represent exons and introns, respectively) and *ZmWAKL* cds on the corresponding schematic diagrams. Please refer to the revised Figure 1 (panel 1b and 1i).

Reviewer #2:

Remarks to the Author:

GLS is a destructive disease on maize. And very few resistant genes against GLS were cloned. This manuscript described the cloning of ZmWAKL, a resistant gene against GLS by map-based cloning. Based on the disease evaluation of transgenic plants, it was confirmed that ZmWAKL gene can enhance maize resistance against GLS. Further, the authors uncovered the ZmWAKL-ZmWIK-ZmBLK1-ZmRBOH4 module which mediates resistance against GLS. This work is excellent.

Response: My co-authors and I would like to express our gratitude to you for reading and evaluating this work. We'll carefully consider and implement your thoughtful comments and constructive suggestions to improve the quality of the manuscript. Thank you again for your time and feedback.

Major comments:

1. No solid evidences were provided to confirm the genetic relationships among ZmWAKL, ZmWIK, ZmBLK1 and ZmRBOH4 in maize plants. Although the authors did a lot of *in vitro* phosphorylation assays and SLC assays, but those results are not enough to confirm their genetic relationship in maize plants. I strongly suggest the authors do some crosses by using transgenic maize plants and mutants to examine the relationship among the four resistant genes. For example, the authors can examine the relationship between ZmWAKL and ZmWIK by checking the disease phenotypes of ZmWAKL/*zmwik*, Ubi::ZmWIK/*zmwkl*, ZmWAKL, *zmwik* mutant and transgenic wild type plants. Also, the authors mentioned that ZmWAKL and ZmWIK phosphorylate each other (Line 321). This means that ZmWIK might work on the upstream of ZmWAKL protein. So, it is required to confirm the relationship between ZmWAKL and ZmWIK, between ZmWIK and ZmBLK1, between ZmBLK1 and ZmRBOH4.

Response: Thanks for your comments. We fully agree with your opinions/suggestions regarding understanding the genetic relationship among ZmWAKL, ZmWIK, ZmBLK1 and ZmRBOH4. Your insights have been valuable in advancing our understanding of these genetic interactions. This project has been conducted for more than 10 years, and it has taken a long time to generate transgenic events for different candidate genes discovered at different times. Currently, we have already started making crosses among the wild-type, mutants, and transgenic lines. We estimate that it will take about 5 years to complete the crosses and to assess their GLS resistance in the field tests. In this manuscript, we have investigated the protein-protein interplay and transphosphorylation in detail to confirm the relationships among these four proteins. Furthermore, we have successfully revealed the genetic effect of each causative gene on GLS resistance. Additionally, in the revised version, we have confirmed the phosphorylation signaling *in vivo* in the four-protein immune module (Fig. 6c). Taken together, these data provide an overview of ZmWAKL-mediated immune response to GLS disease.

Regarding the receptor ZmWAKL^Y and its co-receptor ZmWIK, the reviewer raises a very interesting question concerning their genetic relationship and position in the four-protein immune module. Our data show that ZmWAKL^Y and ZmWIK can phosphorylate each other, but they are not equivalent, with ZmWIK being more strongly phosphorylated than ZmWAKL^Y. During the functional validation of ZmWIK, we obtained an EMS-induced loss-of-function mutant *Zmwik* in the B73 background, which harbors a premature translation termination codon. The mutant was backcrossed twice to NIL-Y32 and self-pollinated to introduce the disease-resistant ZmWAKL^Y gene into the homozygous *Zmwik* mutant to assess GLS resistance (Fig. 3l-n). Through molecular marker detection, we found that both ZmWIK wild-type and its *Zmwik* mutant carry ZmWAKL^Y (Fig. R1). In this case, the ZmWAKL^Y/*Zmwik* plants significantly reduced maize resistance to GLS compared to ZmWAKL^Y/ZmWIK plants (Fig. 3n). Taken together, ZmWIK must be in the same signaling pathway as ZmWAKL^Y and acts downstream of ZmWAKL^Y in GLS resistance. We have modified description in the text, for details, please refer to lines 332-337.

Fig R1. The genotypes of the *qRgls1* locus for the wild-type *ZmWIK* and its *Zmwik* mutant. Gel electrophoresis images of PCR products using different molecular markers. NIL-Y32 contains the disease-resistant *ZmWAKL^Y* allele and serves as a positive control, while NIL-Q11 carried the susceptible *ZmWAKL^O* allele and serves as a negative control. The *GI90* is a dominant marker, with bands indicating the presence of the resistance *ZmWAKL^Y* allele and no bands meaning the absence of the resistance *ZmWAKL^Y* allele (or the presence of the susceptible *ZmWAKL^O* allele). The *35-5-3* is a co-dominant marker, and when the amplified band size matches the positive control, it indicates the presence of the resistance *ZmWAKL^Y* allele, while matching the negative control means the absence of the resistance *ZmWAKL^Y* allele (or the presence of the susceptible *ZmWAKL^O* allele). These results indicate that both the wild type *ZmWIK* and its *Zmwik* mutant carry the disease-resistant *ZmWAKL^Y* allele.

2. Since phosphorylation of *ZmWAKL*, *ZmWIK*, *ZmBLK1* and *ZmRBOH4* are critical for the signal transduction, I strongly suggest the authors check whether the phosphorylation levels of those proteins are induced in transgenic maize plants by *C. zeina* infection. Also, it is better to check whether the enhanced phosphorylation levels of *ZmBLK1* and *ZmRBOH4* induced by *C. zeina* infection are dependent on *ZmWAKL^Y* by using transgenic plants.

Response: Thanks for the constructive suggestions. We are also interested in revealing the changes in phosphorylation levels of the four proteins after *C. zeina* infection. Using transgenic plants overexpressing *ZmWAKL^Y-GFP*, we did observe increased phosphorylation levels after *C. zeina* inoculation, peaking at 6 hours (Fig 6a). Conversely, transgenic plants overexpressing *ZmWAKL^O-GFP* did not show any change in the phosphorylation levels in response to *C. zeina* infection (Fig 6b). As for the other three proteins, both tagged transgenic plants and specific phosphorylated antibodies are currently not available to directly assess their phosphorylation levels. This is why we did not include the phosphorylation levels of the three proteins in our initial submitted manuscript. Instead, in the revised version, we performed mass spectrometry-based phosphoproteomics to search for *C. zeina*-induced genome-wide phosphorylation changes. To achieve this, we infected the transgenic plants overexpressing *ZmWAKL^Y-GFP* and *ZmWAKL^O-GFP* at their 4-leaf seedling stages with *C. zeina* and collected leaf tissues at 6 hpi for total protein extraction. The proteins were subjected to in-gel digestion and mass spectrometry analysis. However, we failed to detect *ZmWIK*, probably because it is a trans-membrane protein. For *ZmBLK1* and *ZmRBOH4*, the phosphorylation levels did show significant increases in *ZmWAKL^Y-GFP* plants compared to *ZmWAKL^O-GFP* plants. This further suggests that the phosphorylation signaling is initiated from *ZmWAKL^Y* upon perceiving pathogen invasion and eventually converges to *ZmRBOH4*. We hope that all these results could provide a “Yes” answer to your question regarding “whether the enhanced phosphorylation levels of *ZmBLK1* and *ZmRBOH4* induced by *C. zeina* infection are dependent on *ZmWAKL^Y*”. We have added these data in the revised version. For specific details, please refer to lines 481-489.

3. Does introgression of *ZmWAKL^Y* allele into susceptible maize lines affect their yield in GLS region and non-GLS region?

Response: That's a very good question. It is known that crop resistance is related to yield in very complicated manners. To explore how the *ZmWAKL^Y* allele affect the yield with or without *C. zeina*, we first integrated the resistant *qRgls1* locus into a GLS-susceptible line PH4CV via MAS to obtain the converted line PH4CV^{*qRgls1*} (Fig. R2a). We found that PH4CV^{*qRgls1*} showed significantly enhanced GLS resistance (Fig. R2a, b). Next, we analyzed whether *qRgls1* affect the yield-related traits. The results show that the introgression of *qRgls1* does not affect the 100-seed weight and ear weight in non-GLS conditions (Fig R2c, d). However, in GLS-pandemic region, the 100-seed weight was significantly higher in PH4CV^{*qRgls1*} than in PH4CV (Fig. R2c), but not the ear weight (Fig R2d). Since we have only estimated the agronomic traits for the inbred lines, the results are very preliminary and not enough to reveal the genetic effect of *ZmWAKL^Y* on yield. Next, we will use PH4CV^{*qRgls1*} and PH4CV to separately cross with other inbred lines to produce F₁ hybrids. Under both GLS and non-GLS conditions, we will elaborately estimate the genetic effects of the *ZmWAKL^Y* allele on yield and yield-related traits.

Fig R2. The phenotypes of inbred line PH4CV and its converted line PH4CV^{*qRgls1*}
a,b, GLS symptoms (a) and scales (b) of PH4CV and PH4CV^{*qRgls1*};
c,d, The performances of PH4CV and PH4CV^{*qRgls1*} in terms of 100-seed weight (c) and ear weight (d) under normal field and disease nursery conditions.
 Statistical significance was determined using a two-sided Student's *t*-test. ns, not significant.

4. Scientific writing should be comprehensively improved in general.

Response: Thanks! As suggested, we have thoroughly revised our manuscript and have also asked a native speaker to further revise the MS for improved readability.

5. Since GLS can be caused by *C. zeina* and *C. zea-maydis*, did the authors test whether *ZmWAKL* contributes resistance against *C. zea-maydis*?

Response: Thank you very much for the comment. In Northeast China, GLS is caused by *C. zea-maydis* (Du, L., Yu, F., Zhang, H. et al, 2020, Genetic mapping of quantitative trait loci and a major locus for resistance to grey leaf spot in maize. *Theor Appl Genet* 133:2521–2533. <https://doi.org/10.1007/s00122-020-03614-z>). We used a segregating backcross population derived from a recombinant (Fig. R3a) and two NILs (Fig. R3b) to estimate the genetic effect of *ZmWAKL^Y* on GLS resistance in Liaoning province in Northeast China. We found that plants with *ZmWAKL^Y* showed significantly higher GLS resistance than those without *ZmWAKL^Y* (Fig. R3a, b). In the current year and subsequent years, we will use all transgenic lines and mutants involved in the

ZmWAKL-ZmWIK-ZmBLK1-ZmRBOH4 immune module to investigate the genetic effects of the relevant genes in resistance to *C. zea-maydis* in Northeast China.

Using the method described previously (Korsman J, Meisel B, Kloppers F, Crampton B, Berger D, 2012, Quantitative phenotyping of grey leaf spot disease in maize using real-time PCR. *European Journal of Plant Pathology* 133:461-471. <http://doi.org/10.1007/s10658-011-9920-1>), we not only detected *C. zeina*, but also some *C. zea-maydis* in the fields in Beijing. Considering the distribution of the pathogens and the performance of ZmWAKL-mediated resistance, we speculate that ZmWAKL could confer resistance to both *C. zeina* and *C. zea-maydis*. However, more experimental evidence is required to conclusively address this question.

Fig R3. GLS scale values of the segregating backcross population and the two NILs

a, GLS scales of a segregating backcross population derived from a recombinant. Q11/Q11 represents plants without the resistant *ZmWAKL^Y* allele, while Q11/Q11^{Rgls1} plants contain a single resistant *ZmWAKL^Y* allele. The numerals at the bottom indicate the number of plants.

b, GLS scales of paired NILs. The numerals indicate the number of plants. Statistical significance was determined using a two-sided Student's *t*-test.

6. Line 35, please delete “unidirectional”. No solid evidence were provided to confirm it. The author mentioned that ZmWAKL and ZmWIK phosphorylate each other. So, they regulate each other. In Arabidopsis FLS2-BAK1-BIK1 complex, flg22-activated FLS2 protein phosphorylates BAK1, then BAK1 phosphorylates BIK1. And, BAK1 and FLS2 are substrates of BIK1 (PNAS, 2010, 107:496-501). It is possible the ZmWAKL-ZmWIK-ZmBLK1 complex, similar as FLS2-BAK-BIK1 complex, is complicated and those subunits phosphorylate and regulate each other.

Response: Thanks for your suggestion. We have deleted the term “unidirectional” and modified the relevant descriptions. In the current study, we found that phosphorylation propagation is primarily transduced from ZmWAKL/ZmWIK to ZmBLK1. However, just as mentioned by the reviewer, it is possible that these subunits may phosphorylate and regulate each other, as observed in Arabidopsis.

Reviewer #3:

Remarks to the Author:

The manuscript by Zhong et al describes the fine mapping and cloning of a ZmWAKL gene underlying the major quantitative resistance against gray leaf spot in maize. The WAK gene family has previously identified as quantitative resistant genes in several important crops and thus, these receptors are of high relevance for plant resistance to pathogen infection. The authors discovered the self-association of ZmWAKL and identified its co-receptor WIK. By this, the authors discovered a WAKL/WIK-BLK-RBOH4 phosphorylation cascade and revealed the WAKL transduced ROS production mechanism against *C. Zeina* infection. The molecular experiments results are complete and convincing. Moreover, further genetic evidence is provided by a transgenic assay in maize. This presented work will significantly improve our understanding on quantitative resistant mechanisms and therefore is of high scientific value. The manuscript is well written and all over it was a pleasure to read this work.

Response: We greatly appreciate the reviewer’s positive comments on our work and would like to take this opportunity to thank the reviewer for the constructive suggestions on our manuscript.

There are some minor points I would suggest to modify or include:

1. Are there any variations of WIK, BLK and ZmRBOH4 between Y and Q? In addition to that, several transgenic mutants were made in the B73 genetic background, what’s the difference between B73, Y and Q alleles of these genes

Response: Yes! We did find some variations in ZmWIK, ZmBLK1 and ZmRBOH4 between the parental lines Y32 and Q11 (Fig. R4). As for B73 and Q11, they share the same proteins with the identical amino-acid sequences. For simplicity, we just compare the amino acid sequences of the three proteins between Y32 and Q11. PCR-based sequencing revealed 12 SNPs in the CDS of *ZmWIK* between Y32 and Q11, with two of them resulting in amino-acid changes (Fig. R4a). There are 10 SNPs in the CDS of *ZmBLK1*^Y and *ZmBLK1*^Q, and one of them in the N-terminus leads to an amino-acid change (Fig R.4b). In *ZmRBOH4*^Y and *ZmRBOH4*^Q, we found two amino acid changes and seven indels (Fig. R4c). The variations in these proteins may affect the interaction and signal transduction in the four-protein immune module. This could be the reason why *ZmWAKL* exhibited variable effects under different genetic backgrounds.

Fig. R4. Schematic diagram of ZmWIK, ZmBLK1 and ZmRBOH4

a, b, c, Conserved functional domains and amino acid sequence variations of ZmWIK (a), ZmBLK1 (b) and ZmRBOH4 (c). The superscript “Y” indicates the sequence derived from the resistant parent Y32, and the superscript “Q” means the sequence derived from the susceptible parent Q11.

2. In line 214-215, authors concluded that the induction pattern of WAKL is consistent with the infection process of *C. Zeina*. Is this a part of a not-shown result, or a citation from (missing) references?

Response: Thank you very much for the comment. We apologize for forgetting to add the source of the result in the MS. In the revised version, we have referenced the relevant figure (Extended data Fig. 1h). Please refer to lines 230-232.

3. The luciferase signal is not a fluorescence signal. The authors should correct this terminology throughout the paper.

Response: Thank you very much for the correction. We have corrected the term throughout the revised MS.

4. Line 320, the EMS or CRISPR knockout is not a deletion mutant.

Response: Thanks for your suggestion. We changed “deletion” into “null mutation”, please refer to line 338 in the revised version.

5. In line 321-322, the authors need to explain which result support the conclusion of “more frequent phosphorylation is made by ZmWAKLY to ZmWIK”.

Response: Thanks for the comment. When the kinase-active ZmWAKL^{Y/ICD}-MBP and His-ZmWIK^{ICD} were mixed *in vitro*, they enhanced each other's phosphorylation levels, with His-ZmWIK^{ICD} being more strongly phosphorylated than ZmWAKL^{Y/ICD}-MBP (Fig 3i). Additionally, we found that His-ZmWIK^{ICD} could strongly phosphorylate MBP^a, and ZmWAKL^{Y/ICD}-MBP further increased phosphorylation levels of both His-ZmWIK^{ICD} and MBP^a (Fig 3j,k). In contrast, the phosphorylated ZmWAKL^{ICD} bands were almost undetectable (Fig 3j). Based on these results, we speculated that “more frequent phosphorylation is made by ZmWAKL^Y to ZmWIK”. Please refer to lines 320-329 in the revised version.

6. BLK is an important immune signaling component and RBOH4 is crucial for ROS burst. Knock out the such genes could result in the reduction of basal immunity, thus causing the mutant being more susceptible to several disease but not dependent to WAK mediated resistance. The author should include this possibility in the discussion.

Response: Thanks for the comment. We fully agree your deduction regarding such genes. The ROS burst elicits basal responses to many pathogens, and any disruption in either ZmBLK1 or ZmRBOH4 is expected to impair maize immune responses to various diseases. We have included the discussion in lines 587-590. Please refer to the revised MS.

7. As the authors mention in the introduction, WAK can also recognize pathogen effectors. Since the elicitor of WAKL is unknown, the author should not sue “WAK/WAKL-related PTI mechanism” in the line 575.

Response: Thank you very much for the correction, we revised it. Please refer to line 606.

8. In the methods section - what is the concentration of spores of *C. zeina* inoculum used for field assays?

Response: Thanks for the comment. The concentration of spore inoculum used for field assays is 5×10^3 spores/ml. We have appended the spore's concentration in the method section. Please refer to line 791.

Decision Letter, first revision:

16th Oct 2023

Dear Dr. Xu,

Thank you for submitting your revised manuscript "ZmWAKL-mediated immunity underlies quantitative disease resistance to gray leaf spot in maize" (NG-A61841R). It has now been seen by the original referees and their comments are below. The reviewers find that the paper has improved in revision, and therefore we'll be happy in principle to publish it in Nature Genetics, pending minor

revisions to comply with our editorial and formatting guidelines.

Sincerely,
Wei

Wei Li, PhD
Senior Editor
Nature Genetics
New York, NY 10004, USA
www.nature.com/ng

Reviewer #1 (Remarks to the Author):

The authors have addressed all my comments. As I expressed earlier, they have put together a manuscript that clearly shows the role of this RLK in GLS resistance.

I agree with the authors that the proposed genetic experiments by reviewer 2 are beyond the scope of this manuscript.

Reviewer #2 (Remarks to the Author):

The authors have answered all of my questions, and I have no more comments.

Reviewer #3 (Remarks to the Author):

The manuscript by Zhong et al presenta an in-depth analysis of the mechanism of ZmWAKL-mediated quantitative resistance in maize through WAKL/WIK-BLK-RBOH4 phosphorylation cascade. The authors comprehensively addressed the questions point by point and the responses are clear and sound. The manuscript is well written and the results are solid. I have no further question about the manuscript and would like to congratulate the authors to their great achievement.

Author Rebuttal, first revision:

Reviewers' Comments:

Reviewer #1:

Remarks to the Author:

The authors have addressed all my comments. As I expressed earlier, they have put together a manuscript that clearly shows the role of this RLK in GLS resistance.

I agree with the authors that the proposed genetic experiments by reviewer 2 are beyond the scope of this manuscript.

Response: Thank you for the positive feedback on our revised manuscript and for all the inspiring comments.

Reviewer #2:

Remarks to the Author:

The authors have answered all of my questions, and I have no more comments.

Response: We are very pleased to have fully addressed the reviewer's concerns. We also extend our gratitude for the insightful comments from the reviewer, which have helped us to strengthen our conclusions.

Reviewer #3:

Remarks to the Author:

The manuscript by Zhong et al presents an in-depth analysis of the mechanism of ZmWAKL-mediated quantitative resistance in maize through WAKL/WIK-BLK-RBOH4 phosphorylation cascade. The authors comprehensively addressed the questions point by point and the responses are clear and sound. The manuscript is well written and the results are solid. I have no further question about the manuscript and would like to congratulate the authors to their great achievement.

Response: Thanks for the reviewer for his positive comment.

Final Decision Letter:

8th Dec 2023

Dear Dr. Xu,

I am delighted to say that your manuscript "The ZmWAKL-ZmWIK-ZmBLK1-ZmRBOH4 module provides quantitative resistance to gray leaf spot in maize" has been accepted for publication in an upcoming issue of Nature Genetics.

Over the next few weeks, your paper will be copyedited to ensure that it conforms to Nature Genetics style. Once your paper is typeset, you will receive an email with a link to choose the appropriate publishing options for your paper and our Author Services team will be in touch regarding any

additional information that may be required.

Your paper will be published online after we receive your corrections and will appear in print in the next available issue. You can find out your date of online publication by contacting the Nature Press Office (press@nature.com) after sending your e-proof corrections.

Please note that *Nature Genetics* is a Transformative Journal (TJ). Authors may publish their research with us through the traditional subscription access route or make their paper immediately open access through payment of an article-processing charge (APC). Authors will not be required to make a final decision about access to their article until it has been accepted. [Find out more about Transformative Journals](https://www.springernature.com/gp/open-research/transformative-journals)

Authors may need to take specific actions to achieve [compliance](https://www.springernature.com/gp/open-research/funding/policy-compliance-faqs) with funder and institutional open access mandates. If your research is supported by a funder that requires immediate open access (e.g. according to [Plan S principles](https://www.springernature.com/gp/open-research/plan-s-compliance))

then you should select the gold OA route, and we will direct you to the compliant route where possible. For authors selecting the subscription publication route, the journal's standard licensing terms will need to be accepted, including <https://www.nature.com/nature-portfolio/editorial-policies/self-archiving-and-license-to-publish>. Those licensing terms will supersede any other terms that the author or any third party may assert apply to any version of the manuscript.

If you have not already done so, we invite you to upload the step-by-step protocols used in this manuscript to the Protocols Exchange, part of our on-line web resource, natureprotocols.com. If you complete the upload by the time you receive your manuscript proofs, we can insert links in your article that lead directly to the protocol details. Your protocol will be made freely available upon publication of your paper. By participating in natureprotocols.com, you are enabling researchers to more readily reproduce or adapt the methodology you use. [Natureprotocols.com](http://natureprotocols.com) is fully searchable, providing your protocols and paper with increased utility and visibility. Please submit your protocol to <https://protocolexchange.researchsquare.com/>. After entering your [nature.com](http://www.nature.com) username and password you will need to enter your manuscript number (NG-A61841R1). Further information can be found at <https://www.nature.com/nature-portfolio/editorial-policies/reporting-standards#protocols>

Sincerely,
Wei

Wei Li, PhD
Senior Editor
Nature Genetics

New York, NY 10004, USA
www.nature.com/ng